# Beyond Grid-Locked Voxels: Neural Response Functions for Continuous Brain Encoding

**Haomiao Chen**[1,2]**, Keith W. Jamison**[3]**, Mert R. Sabuncu**[1,2,3] **& Amy Kuceyeski**[1,3]
[1]Cornell University, [2]Cornell Tech, [3]Weill Cornell Medicine
`hc872@cornell.edu, kwj2001@med.cornell.edu,`
`msabuncu@cornell.edu, amk2012@med.cornell.edu`

## Abstract

Neural encoding models aim to predict fMRI-measured brain responses to natural images. fMRI data is acquired as a 3D volume of voxels, where each voxel has a defined spatial location in the brain. However, conventional encoding models often flatten this volume into a 1D vector and treat voxel responses as independent outputs. This removes spatial context, discards anatomical information, and ties each model to a subject-specific voxel grid. We introduce the **NRF** Neural Response Function, a framework that models fMRI activity as a continuous function over anatomical space rather than a flat vector of voxels. NRF represents brain activity as a continuous implicit function: given an image and a spatial coordinate $(x, y, z)$ in standardized MNI space, the model predicts the response at that location. This formulation decouples predictions from the training grid, supports querying at arbitrary spatial resolutions, and enables resolution-agnostic analyses. By grounding the model in anatomical space, NRF exploits two key properties of brain responses: (1) **local smoothness**—neighboring voxels exhibit similar response patterns; modeling responses continuously captures these correlations and improves data efficiency, and (2) **cross-subject alignment**—MNI coordinates unify data across individuals, allowing a model pretrained on one subject to be fine-tuned on new subjects. In experiments, NRF outperformed baseline models in both intrasubject encoding and cross-subject adaptation. Achieving high performance while reducing the data size needed by orders of magnitude. To our knowledge, NRF is the first anatomically aware encoding model to move beyond flattened voxels, learning a continuous mapping from images to brain responses in 3D space. Code and project site: https://github.com/haomiao8/NRF

## 1 Introduction

A major goal in computational neuroscience is to understand how the human brain maps visual stimuli into neural activity. Neural encoding models aim to address this by predicting neural responses-typically measured by fMRI—from visual stimuli. These models offer powerful tools for analyzing high-dimensional brain data and probing the representations encoded in the visual system (Downing et al., 2001; Epstein & Kanwisher, 1998; Gu et al., 2022; Heeger & Ress, 2002; Kanwisher et al., 1997; Naselaris et al., 2011; Huth et al., 2012) .

However, the real-world utility of current neural encoding models remains limited. Current neural encoding models represent fMRI responses as a 1D vector in $\mathbb{R}^n$, where n is a subject-specific voxel count (Naselaris et al., 2015; St-Yves & Naselaris, 2018; Wang et al., 2023; Yamins et al., 2014). This discrete formulation has two critical limitations: **1) Ignoring 3D structure**. By flattening fMRI volumes into 1D vectors, conventional models discard spatial information. This removes local context: anatomically adjacent voxels, which are often functionally correlated, are instead treated as independent outputs. **2) Subject specific.** Each model is tied to the voxel grid of a single subject, making it non-transferable across individuals. Because voxel counts differ across brains, the output dimensionality of conventional models and their corresponding architectures are tied to a single subject. As a result, knowledge learned from one individual cannot be directly transferred to

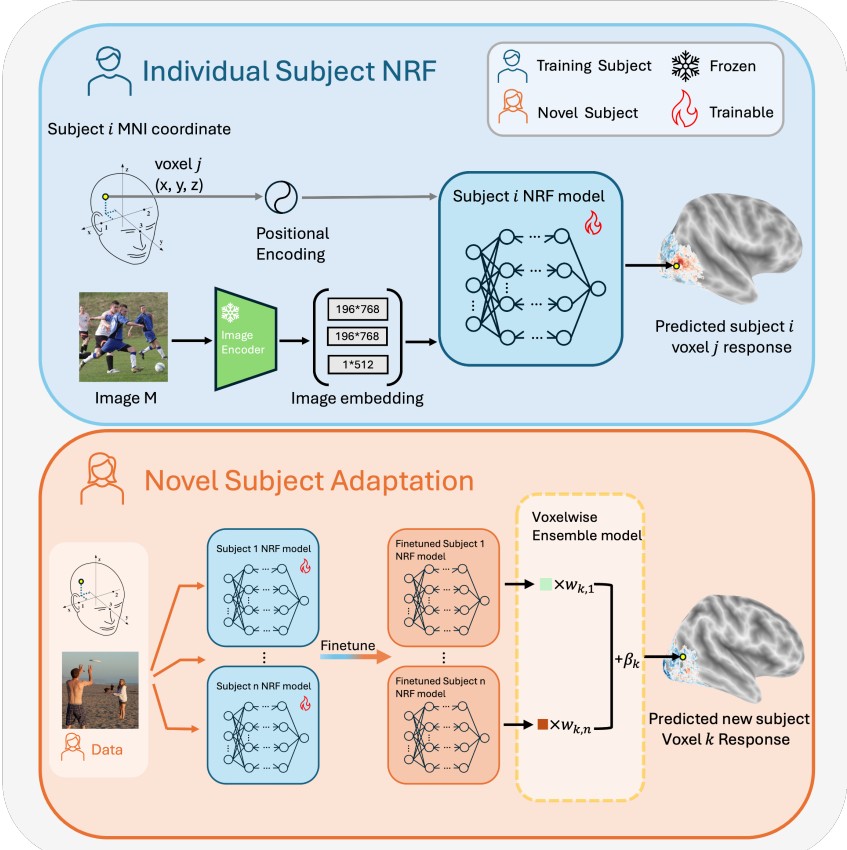

Figure 1: **Overview of NRF. Top:** Individual-subject NRF. Brain responses are modeled as a continuous function of both image features and anatomical coordinates in MNI space. NRF maps image features to voxel responses while capturing correlations between neighboring voxels through shared anatomical coordinates. **Bottom:** New-subject adaptation. For a novel subject, we start from a *pretrained* individual-subject NRF (trained on the other subjects) and then *fine-tune* this model using only a small amount of data from the new subject. The figure shows both the pretrained and the fine-tuned NRFs to make explicit how the base model is adapted during transfer. Predictions from multiple fine-tuned base models are then combined via voxel-wise ensembling to better capture subject-specific variability. NRF thus moves beyond grid-locked voxel models, providing a continuous, anatomically grounded representation that supports both data-efficient single-subject encoding and flexible cross-subject transfer.

another, forcing each new subject to require training a separate model from scratch. Consequently, these models fail to exploit the 3D geometry of brain activity, wasting statistical power and requiring more data to learn accurate mappings.

These issues are particularly problematic in realistic applications, where data are scarce. Unlike large-scale efforts such as NSD (Allen et al., 2022), which collected tens of thousands of trials per subject over a year-long scanning period, most studies can only acquire a few hundred trials per subject due to cost and time constraints. Thus, the inefficiencies of discrete neural encoding models are amplified in real-world low-data regimes. In short, previous encoding models are *grid-locked*: they can only predict responses at the discrete sampling points they were trained on, for a single subject, and at one resolution.

However, the human brain is a continuous 3D structure. Within each subject, neighboring voxels also exhibit similar response patterns, reflecting the spatial smoothness of neural activity. Despite individual variability, the visual cortex is highly conserved across people :areas such as the fusiform face area (FFA) and extrastriate body area (EBA) consistently respond to the same stimulus categories,

and these regions align well across subjects when mapped into standardized anatomical templates such as MNI space (Heeger & Ress, 2002; Kanwisher et al., 1997; Downing et al., 2001; Epstein & Kanwisher, 1998; Heeger & Ress, 2002; Naselaris et al., 2011; Huth et al., 2012). Ignoring this organization—both local smoothness and cross-subject correspondences — discards valuable structure that could enable more efficient learning and better generalization.

To address these limitations, we propose the **Neural Response Function** (NRF), a coordinate-based neural encoding model that predicts fMRI responses as a continuous function over anatomical space. Given a stimulus $M$ and a spatial coordinate $\mathbf{x} = (x, y, z)$ in standardized MNI space, NRF outputs the predicted brain response $r$ at that location:

$$\Phi(M, \mathbf{x}) = r, \mathbf{x} \in \mathbb{R}^3, \ r \in \mathbb{R}.$$

This formulation directly addresses the limitations of previous models:

- **Exploiting local smoothness.** By conditioning predictions on anatomical coordinates, NRF incorporates the 3D spatial structure of fMRI data. This allows nearby voxels that are anatomically connected and functionally correlated to share information instead of being treated as independent outputs. As a result, NRF captures local smoothness in brain responses and achieves greater data efficiency.

- **Efficient cross-subject adaptation.** NRF grounds predictions in standardized MNI space, unifying responses across subjects in a shared coordinate system. This enables direct transfer: a model pretrained on one subject can be adapted to a new individual with only minimal fine-tuning. We further introduce a *finetune–ensemble* strategy that leverages multiple pretrained models to boost adaptation accuracy, reducing the need for extensive subject-specific data collection in real-world settings.

- **Resolution-agnostic modeling.** By defining responses in continuous space, NRF decouples predictions from the specific voxel grid used during training. The model can be queried at arbitrary spatial coordinates, independent of the voxel size or sampling scheme used during data collection. This allows for the seamless integration of data acquired at varying resolutions and opens the door to building general-purpose brain models that move closer to a functional *digital twin* of the human visual system.

Through experiments,we demonstrate that NRF provides a novel anatomy-grounded framework for neural encoding, offering a new paradigm for efficient and generalizable brain modeling.

## 2 RELATED WORK

**Neural encoding models.** fMRI encoding models have been extensively studied over the past two decades (Mitchell et al., 2008; Huth et al., 2016; Gu et al., 2022; Tang et al., 2023; Kay et al., 2008; Güçlü & Van Gerven, 2015; Naselaris et al., 2015). Most existing approaches treat fMRI data as discrete, formulating the problem as a regression task that maps image features to voxel-wise responses (Naselaris et al., 2011; Han et al., 2019; Wang et al., 2023).Recent work has also explored stronger transformer-based encoders (Adeli et al., 2025; Beliy et al., 2024). In these models, fMRI responses are flattened as a 1-D vector, and each voxel is treated independently, ignoring the 3D anatomical structure of the brain. While some approaches incorporate spatial priors, such as fitting multi-parameter models for spatial frequency mapping (Broderick et al., 2022) or using Bayesian templates to warp retinotopic maps (Benson & Winawer, 2018). However, general encoding frameworks largely fail to exploit the local smoothness and cross-subject correspondences inherent in brain activity. To our knowledge, NRF is one of the first anatomically aware encoding models that formulates image to fMRI prediction mapping as a continuous function over 3D brain space, leveraging anatomical structure to improve data efficiency and generalization.

**Neural decoding models.** A parallel line of work focuses on reconstructing or decoding visual stimuli from fMRI signals. Early approaches (Scotti et al., 2024) typically utilize subject-specific MLPs that cannot naturally generalize across individuals due to varying voxel counts and layouts. Recent methods address this via adaptive pooling for cross-subject decoding (Wang et al., 2024) or by incorporating voxel (x,y,z) coordinates into attention mechanisms via positional encodings (Qiu et al., 2025). While these methods highlight the importance of anatomical alignment, they

pursue a fundamentally different goal: decoding information from measured, discrete voxel grids. Even with spatial cues, their operations remain tied to the specific voxels acquired. In contrast, we address the encoding problem by treating the brain as a continuous 3D structure. Rather than using coordinates as auxiliary features for discrete voxels, NRF models neural activity as a continuous function in brain space. This formulation enables resolution-agnostic querying at arbitrary positions and facilitates seamless cross-subject adaptation within a unified anatomical coordinate system.

**Brain semantic mapping.** A rich line of research has mapped the semantic organization of the cortex using voxel-wise encoding models (Huth et al., 2016; Deniz et al., 2019). These studies typically train independent encoders on flattened 1D response vectors and project learned selectivities onto the cortical surface post hoc for visualization. While this pipeline uses 3D spatial information during analysis, the encoding models themselves do not incorporate anatomical structure during training; each voxel is modeled independently, ignoring its physical location. NRF differs by injecting anatomical structure directly into the learning process. Rather than using coordinates solely for downstream visualization, NRF conditions the encoding function on 3D $(x, y, z)$ coordinates and learns a *continuous* neural field over standardized brain space. This allows the model to operationalize the spatial principles revealed in prior studies, such as the fact that nearby voxels exhibit smoothly varying semantic selectivity. By modeling neural activity as a spatially smooth function rather than a set of independent points, NRF achieves greater data efficiency and enables flexible cross-subject adaptation without the need to resample volumes onto a shared grid.

**Implicit neural representation** Implicit neural representations(INR) have emerged as a powerful paradigm for modeling continuous signals in computer vision and graphics. Instead of storing data on fixed grids, INRs represent signals such as images (Sitzmann et al., 2020) and 3D shapes (Park et al., 2019; Mildenhall et al., 2021; Chen & Zhang, 2019; Mescheder et al., 2019) as continuous functions parameterized by neural networks. A key advantage of this framework is its ability to capture fine-grained structure and support resolution-agnostic queries. Inspired by this line of work, we adopt a similar coordinate-based formulation for fMRI encoding. Unlike prior voxel-wise models that discretize the brain into subject-specific grids, our approach treats brain responses as a continuous function over standardized anatomical coordinates. To our knowledge, NRF is the first attempt to bring the implicit representation framework to computational neuroscience, enabling anatomically aware, resolution-agnostic modeling of fMRI responses.

## 3 METHOD

### 3.1 MODELING NEURAL RESPONSE AS IMPLICIT NEURAL REPRESENTATION

Current encoding models can be summarized in two steps: flatten neural response into 1D vectors, then train an encoding model that takes an image or its embedding as input and directly outputs the predicted response as a flattened vector in $\mathbb{R}^n$. Ignoring the 3D spatial information and forcing models to be trained separately for each subject. This leads to poor data efficiency. Our key insight is that brain response should be modeled in its anatomical context. We represent the brain response mapping as a **continuous function** over MNI coordinates, a standardized anatomical space. Formally, given a stimulus image $M$ and a spatial coordinate $\mathbf{x} = (x, y, z) \in \mathbb{R}^3$, the Neural Response Function (NRF) outputs the predicted fMRI response $\hat{r} \in \mathbb{R}$ at that location:

$$\Phi(M, \mathbf{x}) = \hat{r}.$$

Rather than outputting a fixed-length vector tied to a particular subject's voxel grid, NRF predicts the response at any coordinate $(x, y, z) \in \mathbb{R}^3$. This shift from 1D discrete outputs to 3D spatial coordinate conditioned continuous predictions makes the model anatomically aware and able to exploit spatial smoothness during training and inference. Because $\Phi$ is defined over $\mathbb{R}^3$, it can be queried at arbitrary spatial resolutions, independent of the voxel grid or sampling scheme used during acquisition. This enables flexible data analysis: fMRI responses can be resampled seamlessly at different resolutions, supporting resolution-agnostic modeling and analysis.

**Intuition.** The continuous module $\Phi$ of NRF learns a displacement from MNI coordinates into a *functional response space*: each anatomical location $\mathbf{x}$ is mapped to an embedding that characterizes

the stimulus features it responds to. The predicted scalar response $\hat{r}$ then arises from comparing this functional embedding against the representation of the stimulus M. Under this view, $\Phi$ constructs a continuous functional atlas, where anatomy provides the substrate and the learned displacement carries the tuning. This factorization of anatomical position and functional identity is what enables a single model to share parameters across subjects despite their differing voxel grids.

**Architecture.** We instantiate $\Phi$ using a two-component design. The first component, $G$, the image feature extraction block. It extracts multi-scale features from the stimulus image $M$, capturing both low-level and high-level representations. These features are fused together to obtain a final image embedding $G(M)$. The second component is an implicit neural representation predictor $P$ that conditions on both $G(M)$ and the spatial coordinate $\mathbf{x}$. The coordinate is first encoded using Fourier features (Tancik et al., 2020):

$$\gamma(\mathbf{x}) = [\cos(2\pi b_1^T \mathbf{x}), \sin(2\pi b_1^T \mathbf{x}), \ldots, \cos(2\pi b_m^T \mathbf{x}), \sin(2\pi b_m^T \mathbf{x})]^T,$$

where $b_j$ are sampled from an isotropic Gaussian. Finally, $G(M)$ and $\gamma(\mathbf{x})$ are concatenated and passed through an MLP predictor $P$:

$$\Phi(M, \mathbf{x}) = P(G(M), \gamma(\mathbf{x})).$$

This design makes the mapping explicitly anatomy-aware by conditioning on both image content and spatial location. Details can be found in the Appendix A.1.

**Model Training.** The model is trained end-to-end with all components learned together, where the objective of the model is to correctly predict the voxel activation on each input image. Training batches are constructed from 32 randomly selected images, where for each image, we randomly sample 2000 voxels (out of 13000-15000 voxels) along with their corresponding fMRI activations for prediction. The model is trained using the Adam optimizer with a learning rate of 3e-3. For the loss function, we employ the same loss as in (Beliy et al., 2019), a convex combination of mean square error and cosine similarity between the predicted response $\hat{r}$ and ground truth fMRI, $r$. The fMRI loss is defined as:

$$L(\hat{r}, r) = (1 - \alpha)\|\hat{r}, r\|_2 + \alpha * cos(\angle(\hat{r}, r))$$

Where $\alpha$ is set to 0.1 during training, which balances absolute error minimization (via MSE) with representational alignment (via cosine similarity).

## 3.2 CROSS SUBJECT TRANSFER

A major challenge in training visual encoding models is the limited availability of subject-specific data. Collecting fMRI responses for thousands of images requires many hours of scanning, often across multiple sessions, and is infeasible in most clinical or experimental settings. In practice, new subjects often contribute only a few hundred trials. Discrete neural encoding models underperform in this regime because they are tied to subject-specific voxel grids: each subject requires training a new model from scratch, and knowledge cannot be transferred directly across individuals.

NRF overcomes this limitation by being **voxel-grid agnostic**. Since responses are defined as a continuous function over standardized MNI space, subjects are naturally aligned in a shared anatomical coordinate system. This enables direct transfer: a model trained on one subject can be adapted to another without voxel-wise resampling, on the new subject's coordinates and responses. Unlike classical voxel-wise models—which rigidly tie the representation to a subject-specific grid—NRF learns a continuous, anatomically grounded representation that could flexibly generalize across individuals with only minimal data. To exploit this property, we adopt a two-step adaptation strategy:

**Finetuning.** A pretrained NRF is fine-tuned on the new subject's limited data, using their MNI coordinates and measured responses. The two components of NRF $\Phi$ are the feature extractor $G$ and MLP predictor $P$. $G$ encodes the visual stimulus into a representation, while $P$ maps this representation and the spatial coordinate to the predicted brain response. Both $G$ and $P$ benefit from adaptation, since individuals vary in both how visual content is processed and how it is mapped to anatomy. Therefore, we perform full end-to-end finetuning of both components on the new subject's data.

**voxel-wise Ensemble.** To further improve performance on new subjects, we perform a voxel-wise ensemble of the predictions from different finetuned models. Similar to the personalized ensemble approach in (Gu et al., 2022), this strategy maximizes predictive performance while preserving inter-subject variability to improve model personalization. Specifically, for each voxel $v$, let $\{\hat{r}_v^{(1,i)}, \hat{r}_v^{(2,i)}, \ldots, \hat{r}_v^{(K,i)}\}$ denote the predictions of $K$ finetuned base models for the $i$th image. We then learn voxel-specific weights $w_{v,k}$ (one per base model, for the $k$ th base model) and a bias $b_v$ by solving a least-squares regression on the limited new subject training data:

$$\min_{\{w_{v,k}, b_v\}} \sum_{i=1}^{N} \left( r_v^{(i)} - \sum_{k=1}^{K} w_{v,k}\, \hat{r}_v^{(k,i)} - b_v \right)^2, \tag{1}$$

where $r_v^{(i)}$ is the measured response of voxel $v$ to image $i$, and $N$ is the number of adaptation samples. At inference time, the final prediction for voxel $v$ is given by the weighted ensemble:

$$\hat{r}_v = \sum_{k=1}^{K} w_{v,k}\, \hat{r}_v^{(k)} + b_v. \tag{2}$$

This ensemble leverages common neural structure while accounting for subject-specific variability, yielding higher accuracy in the low-data regime and producing a more personalized model for each subject.

## 4 EXPERIMENTAL SETUP

### 4.1 DATASETS AND PREPROCESSING

We use the Natural Scene Dataset (NSD) (Allen et al., 2022), which includes whole-brain 7T fMRI data from 8 subjects who viewed 10,000 natural scene images from the MS COCO dataset, repeated 1-3 times. The brain activations were computed using the GLMSingle algorithm (Prince et al., 2022), and each voxel's response value is normalized per session ($\mu = 0, \sigma^2 = 1$). The brain activation to repeated images within a subject was averaged. The Neural response function(NRF) was trained using 9,000 unique images per subject, with around 1,000 images used for testing model accuracy via voxel-wise pearson correlation ($r$). Since we are focusing on the visual cortex regions, we apply the official nsdgeneral region-of-interest (ROI) mask, which spans visual regions ranging from the early visual cortex to higher visual areas. Our evaluation focuses on Subj01, Subj02, Subj05, and Subj07 because these subjects completed all experiment sessions.

### 4.2 EVALUATION METRICS.

To quantitatively compare with other models, we assess model performance across two levels.

**Voxel-Level Metrics:** To quantify prediction accuracy, we compute the voxel-wise Pearson correlation ($r$) and voxel-wise mean square error (MSE) across all testing images.

**Semantic-Level Metrics.** Following prior work (Bao et al., 2025), we evaluate the semantic fidelity of predicted responses using MindEye2 (Scotti et al., 2024), a pretrained fMRI-to-image decoder. Given an input image, we first predict its fMRI response using NRF; the predicted response is then fed into MindEye2, which reconstructs the corresponding visual stimulus. We compare these reconstructions against the ground-truth images presented during data collection.We employ a suite of image reconstruction metrics. PixCorr and SSIM quantify low-level visual fidelity, while Alex(2) and Alex(5) measure similarity in early and deeper layers of AlexNet. To evaluate higher-level semantic alignment, we compute Incep, CLIP, Eff, and SwAV scores, which assess the representational correspondence between reconstructed and original images in diverse semantic embedding spaces. Additional details are provided in Appendix A.3.

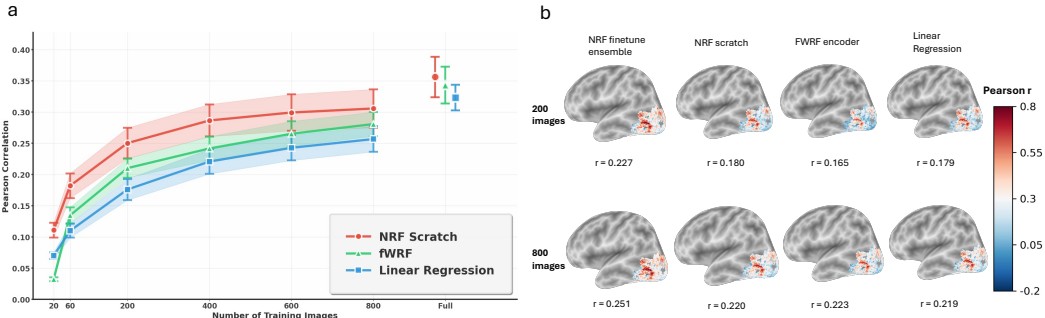

Figure 2: Prediction accuracy (Pearson correlation $r$) in low data regime. a. **Single-subject models.** NRF consistently outperforms baseline models when trained on limited samples from scratch, highlighting the benefit of its continuous mapping.Results are shown for the **mean of the median voxel correlation across four subjects**, with error bars indicating the **standard error of the mean (SEM)**. b. **Cross-subject transfer.** Voxel-level prediction accuracy visualized on the cortical surface of subject 7. When pretrained base models from other subjects are available, the NRF finetune ensemble further improves performance over NRF scratch and baselines, showing clear gains across visual regions.

## 5    RESULTS

### 5.1    INDIVIDUAL SUBJECTS ENCODING

We first evaluated NRF's neural prediction capability for single-subject data. Training a separate model for each of the 4 subjects and comparing the average neural prediction accuracy across subjects. For comparison, we selected two representative encoding models as baseline comparison. The linear regression model from the BrainDIVE (Luo et al., 2023) and the fWRF (Feature-Weighted Receptive Field) encoder (St-Yves & Naselaris, 2018). Details about the baseline model are in Appendix A.2. We also took the result for full data encoding performance from MindSimulator (Bao et al., 2025).

We first evaluate NRF under limited-data conditions, since practical applications rarely have access to the tens of thousands of trials collected in large-scale datasets such as NSD. As shown in Figure 2a, NRF achieves significantly higher accuracy than baseline models when trained on small numbers of images. Remarkably, with only 200 training samples, NRF outperforms baselines trained on more than 800 images. We attribute this data efficiency to the anatomical awareness of NRF: by conditioning on spatial coordinates, the model can exploit the smoothness of fMRI responses and learn more effectively from scarce data. This neuroscience-inspired design makes NRF particularly well-suited for realistic, low-data regimes.

Next, we evaluate NRF in the full-data setting(∼9k training images). Quantitative results, summarized in Table 1, show that NRF outperforms baselines on voxel-wise prediction metrics while achieving comparable performance on semantic-level evaluations. In addition, we observed that some baselines, such as fWRF, achieve unusually high semantic-level scores, in some cases even surpassing reconstructions from measured fMRI. We attribute this to decoder bias: fWRF outputs, while less neurally accurate, may align more closely with the pretrained decoder's distribution, thereby inflating semantic metrics. These results indicate that semantic-level metrics should be interpreted as a coarse indication of reconstruction quality rather than a strict basis for comparing encoding models. Further discussion regarding these metrics is included in Appendix A.3.

Figure 3 provides a qualitative assessment via image reconstructions; the decoded images capture low-level visual features and high-level semantic categories with high fidelity to the ground truth. These results demonstrate that NRF maintains high voxel-level accuracy while also preserving semantic information, confirming its effectiveness across both limited and full data regimes. Appendix A.4 presents detailed per-subject results along with the voxel-wise accuracy distributions, and Appendix A.5 reports the statistical significance analyses.

| Method | Voxel-Level | | Semantic-Level (via decoding) | | | | | | | |
|---|---|---|---|---|---|---|---|---|---|---|
| | Pearson↑ | MSE↓ | PixCorr↑ | SSIM↑ | Alex(2)↑ | Alex(5)↑ | IncepT↑ | CLIP↑ | Eff↓ | SwAV↓ |
| Measured fMRI | – | – | 0.322 | 0.431 | 96.1% | 98.6% | 95.4% | 93.0% | 0.619 | 0.344 |
| Linear Regression | 0.323 | 0.353 | 0.186 | 0.271 | 86.1% | 95.0% | 90.2% | 84.5% | 0.750 | 0.417 |
| fWRF | 0.343 | 0.361 | 0.303 | 0.341 | 96.9% | 99.1% | 96.2% | 91.9% | 0.614 | 0.356 |
| MindSimulator (Trials=1) | 0.345 | 0.403 | 0.194 | 0.296 | 89.0% | 96.2% | 92.3% | 90.3% | 0.702 | 0.399 |
| MindSimulator (Trials=5) | 0.355 | 0.385 | 0.201 | 0.298 | 89.6% | 96.8% | 93.2% | 91.2% | 0.688 | 0.393 |
| NRF (our method) | **0.358** | **0.345** | 0.261 | **0.371** | 91.6% | 96.3% | 92.1% | 89.3% | 0.706 | 0.400 |

Table 1: Evaluation results of fMRI prediction accuracy for the model trained on the full dataset. All reported voxel-level metric values are reported as the **per-subject median across voxels**, and the table shows the **mean of these medians across the 4 subjects**. Additional subject-wise results and full per-voxel distributions are provided in Appendix A.4.

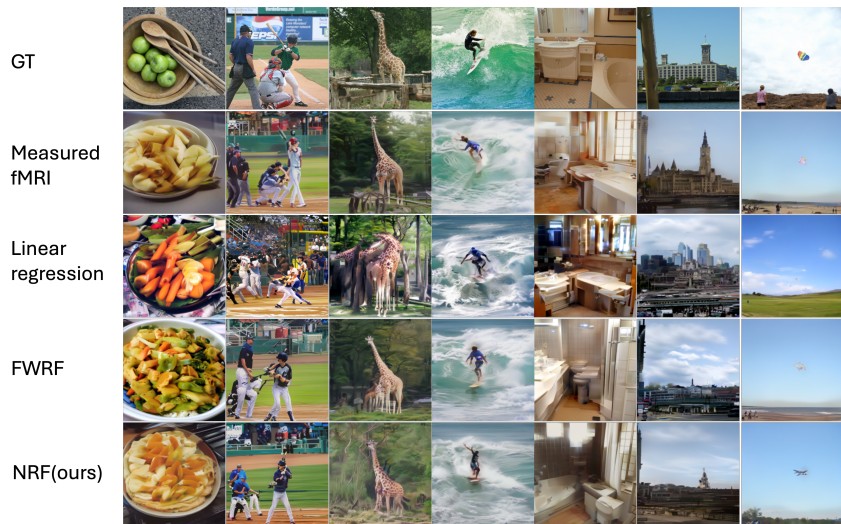

Figure 3: Visualization comparison between different neural encoding models and NRF. GT = seen during data collection. Measured fMRI = decoded image using measured fMRI. Reconstructions from NRF-predicted responses preserve both low-level visual details and high-level semantic content of the stimuli. Results shown for Subject 1.

## 5.2 NEW SUBJECT ADAPTATION

More importantly, NRF enables cross-subject transfer, allowing knowledge learned from one subject to be adapted to new subjects—a critical property given that collecting fMRI data for new individuals is both resource-intensive and time-consuming. To evaluate this capability, we tested adaptation with 20, 200, and 800 images, corresponding to approximately 4, 40, and 160 minutes of scanning time. Three subjects were used for pretraining base models, and a fourth subject was held out for adaptation. For the new subject, we applied fine-tuning followed by voxel-wise regression ensemble using the limited data. As a baseline, we compared against the "NRF scratch" approach, where a new NRF is trained entirely from the same limited dataset without pretraining. Across all data conditions, fine-tuning + ensemble consistently outperformed NRF scratch, confirming that NRF's anatomically grounded formulation enables efficient cross-subject transfer, reducing the need for extensive subject-specific data while maintaining high predictive fidelity. The qualitative comparison is shown in Table 2. Prediction accuracy comparison across different methods is shown in Figure 2b. Notably, in the very low-data regime, finetuning + ensemble achieved strong semantic-level decoding performance. This shows that the strategy not only improves voxel-wise prediction but also preserves subject variability, enabling predicted responses that more faithfully capture the semantic content of visual stimuli.

| Training Images | Method | Voxel-Level | | Semantic-Level (via decoding) | | | | | | | |
|---|---|---|---|---|---|---|---|---|---|---|---|
| | | Pearson↑ | MSE↓ | PixCorr↑ | SSIM↑ | Alex(2)↑ | Alex(5)↑ | IncepT↑ | CLIP↑ | Eff↓ | SwAV↓ |
| Full | NRF subject 7 (all data) | 0.269 | 0.348 | 0.244 | 0.367 | 0.880 | 0.936 | 0.892 | 0.846 | 0.768 | 0.445 |
| 20 | NRF scratch | 0.076 | **0.417** | 0.060 | 0.195 | 0.564 | 0.597 | 0.549 | 0.545 | 0.962 | 0.621 |
| | NRF finetune ensemble | **0.114** | 0.445 | **0.186** | **0.366** | **0.750** | **0.792** | **0.732** | **0.729** | **0.868** | **0.515** |
| 200 | NRF scratch | 0.180 | 0.394 | 0.159 | 0.284 | 0.760 | 0.813 | 0.774 | 0.716 | 0.857 | 0.515 |
| | NRF finetune ensemble | **0.227** | **0.390** | **0.255** | **0.372** | **0.908** | **0.957** | **0.913** | **0.873** | **0.729** | **0.425** |
| 800 | NRF scratch | 0.220 | 0.376 | 0.188 | 0.313 | 0.856 | 0.926 | 0.878 | 0.834 | 0.772 | 0.452 |
| | NRF finetune ensemble | **0.251** | **0.372** | **0.269** | **0.382** | **0.927** | **0.970** | **0.922** | **0.895** | **0.700** | **0.408** |

Table 2: New-subject adaptation with limited data (20, 200, 800 images). Results are shown for adapting the NRF pretrained on subjects 1, 2, and 5 to subject 7 using finetuning + ensemble. Performance is reported as the **median across all voxels of subject 7** for voxel-level metrics. The adapted NRF consistently outperforms training from scratch, and with only 200 images, it even exceeds the performance of a model trained on the full dataset.

## 5.3 PROBING ANATOMICAL AWARENESS

To verify the mechanisms behind NRF's performance, particularly in data-constrained settings, we probe its reliance on two fundamental properties of fMRI data: (i) the *local spatial continuity* of voxel responses within a subject, and (ii) the *anatomical alignment* across subjects. We conduct controlled ablation experiments by introducing structural perturbations that systematically disrupt these factors. If NRF's efficiency stems from its anatomical grounding, performance should degrade significantly when these structural priors are violated.

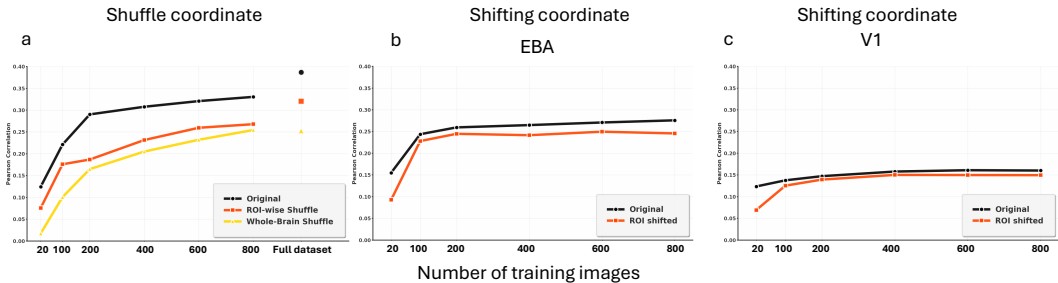

Figure 4: Probing anatomical awareness in NRF. (a) Disrupting spatial smoothness by shuffling coordinate–response pairings reduced accuracy, especially in low-data regimes, confirming that NRF relies on local continuity in brain responses. (b)(c) Breaking cross-subject alignment by shifting MNI coordinates degraded transfer, with the largest effect under limited data, showing that anatomical correspondence is critical for efficient adaptation.

**Disrupting Local Smoothness via Voxel Shuffling.** To test whether NRF's data efficiency stems from exploiting spatial continuity, we disrupted the natural smoothness of fMRI data by shuffling coordinate–response pairings. Voxel responses were randomly reassigned to MNI coordinates, breaking correlations between neighboring voxels. We performed two variants of this perturbation: (i) *global shuffling*, randomizing pairings across the entire visual cortex, and (ii) *ROI-wise shuffling*, randomizing only within each ROI. Traditional voxel-wise models should be unaffected, since they treat voxels independently. In contrast, NRF relies on coordinate conditioning, and as expected, its performance dropped sharply in low-data regimes, with global shuffling producing the largest drop. This confirms that NRF's improvements are driven by its ability to leverage local smoothness in brain responses. Shown in Figure 4(a).

**Disrupting Cross-Subject Alignment.** To test the importance of anatomical correspondence for transfer, we disrupted MNI alignment by shifting voxel coordinates between subjects. Specifically, a model pretrained on Subject 1 was finetuned on Subject 7 using responses from EBA and V1. During finetuning, the MNI coordinates were shifted while remaining within the subject's brain range, breaking the cross-subject anatomy alignment. Compared to finetuning with aligned coordinates,

| Method | Voxel-Level | | Semantic-Level (via decoding) | | | | | | | |
|---|---|---|---|---|---|---|---|---|---|---|
| | Pearson↑ | MSE↓ | PixCorr↑ | SSIM↑ | Alex(2)↑ | Alex(5)↑ | IncepT↑ | CLIP↑ | Eff↓ | SwAV↓ |
| NRF finetune ensemble | 0.227 | 0.390 | **0.255** | **0.372** | **0.908** | **0.957** | **91.3%** | **87.3%** | **0.729** | **0.425** |
| NRF finetune average | **0.253** | **0.367** | 0.167 | 0.283 | 0.784 | 0.852 | 80.4% | 74.9% | 0.848 | 0.514 |
| NRF finetune base (subj1→subj7) | 0.220 | 0.386 | 0.246 | 0.375 | 0.897 | 0.952 | 90.8% | 86.9% | 0.735 | 0.431 |
| NRF finetune base (subj2→subj7) | 0.232 | 0.379 | 0.243 | 0.366 | 0.885 | 0.937 | 87.1% | 82.4% | 0.779 | 0.457 |
| NRF finetune base (subj5→subj7) | 0.225 | 0.389 | 0.226 | 0.371 | 0.874 | 0.938 | 87.6% | 82.4% | 0.775 | 0.452 |

Table 3: Ablation on voxel-wise regression ensemble. We report the result for adapting from subjects 1,2,5 to subject 7 with 200 images.

coordinate shifting substantially degraded cross-subject transfer. The effect was most pronounced in low-data regimes: with only a small number of finetuning samples, the misaligned model failed to adapt, whereas alignment enabled effective transfer. With more data, the model gradually compensated for the misalignment, but still required far more samples to match the aligned case. These results demonstrate that NRF's cross-subject generalization depends critically on anatomical alignment. Without it, transfer is possible but far less data-efficient. To avoid artificial overlap after shifting, finetuning was performed using ROI-restricted data rather than the full brain.

## 5.4 ABLATION STUDY

**voxel-wise ensemble** A key component for new subject adaptation is voxel-wise regression ensemble, where each voxel is fit with a linear regression model to optimally combine predictions from multiple fine-tuned base models. This approach improves prediction accuracy while preserving subject-specific variability. Table 3 compares voxel-wise regression against single fine-tuned base models and simple averaging. While simple averaging slightly boosts voxel-wise prediction accuracy, it hinders subject variability and produces predicted fMRI signals with reduced semantic fidelity, leading to lower decoding performance. In contrast, voxel-wise regression leverages complementary information across base models in a flexible, voxel-specific way, achieving both higher voxel-level accuracy and stronger semantic-level decoding results. This suggests that the ensemble does not merely act as a noise-reduction filter but effectively "mixes" specialized knowledge from different source subjects to better represent the unique functional profile of the target individual. Additional ablation results are included in the Appendix A.7.

## 6 CONCLUSION

In this work, we introduced the Neural Response Function (NRF), an anatomically aware neural encoding model that represents fMRI activity as a continuous function over MNI coordinates. Unlike conventional voxel-wise models, NRF leverages spatial smoothness and cross-subject alignment to achieve accurate predictions in low-data regimes and to support efficient subject adaptation. Crucially, its continuous formulation moves beyond grid-locked voxels, allowing predictions at arbitrary spatial resolutions and across individuals. In this sense, NRF serves as a resolution-agnostic digital twin of the brain: a unified, flexible representation that integrates data across scales and subjects. These advances offer a new path toward efficient, generalizable, and anatomically grounded neural encoding.

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

## A  APPENDIX

**Sections**

## A.1 Additional details on NRF

**Image Feature Extraction Block.** We leverage the pretrained OpenAI CLIP ViT-B/16 to obtain multiscale image features. Representations are extracted from the 3rd and 6th transformer layers, each yielding features of shape $(196 \times 768)$, along with the final CLIP embedding of shape $(1 \times 512)$. The two intermediate feature maps are each processed by separate two-layer projection modules with identical architecture: the first layer reduces the dimensionality from $(196 \times 768)$ to $(196 \times 256)$, and the second compresses this to a $(1 \times 256)$ vector. These two compact embeddings are then concatenated with the $(1 \times 512)$ CLIP embedding to form the fused multiscale image representation $G(M)$.

**MLP Predictor.** The predictor is a coordinate-conditioned MLP that takes as input both the Fourier positional encoding of the MNI coordinate and the fused image embedding $G(M)$. It outputs a single scalar—the predicted fMRI response at that voxel location. We use an 8-layer MLP with hidden dimension 4096, applying ReLU activations after each layer except the final output. Further ablations on model architecture are included in Appendix A.7.

## A.2 Additional details on baseline models

**FWRF encoding model** The encoder uses AlexNet as the base feature extractor, processing 227×227 pixel input images normalized to [0,1]. Feature selection retains the top 256 features per layer based on variance across the training data. The receptive field model employs a 3×3 spatial grid with aperture size 0.8, covering RF sizes from 0.15 to 0.25 across 2 logarithmically-spaced scales, yielding 18 total RF candidates per voxel. Ridge regression optimization uses regularization parameters $\lambda \in [10^4, 10^5]$ with adaptive holdout validation.

**Linear Regression Encoding.** For linear regression baseline we used the same encoding model as (Luo et al., 2023). Specifically, we extract the $(1 \times 512)$ CLIP embedding from OpenAI CLIP ViT-B/16 and directly map it to the voxel dimension (e.g., 15,724 voxels) using a linear layer. The model is trained for 150 epochs with the AdamW optimizer, with a learning rate that decays linearly from $3 \times 10^{-4}$. During inference, we select the checkpoint that achieves the lowest validation MSE.

### A.3 Additional Details on evaluation metrics

We used the evaluation metrics for decoded image evaluation from MindEye2 (Scotti et al., 2024) directly. Given an input image, we first predict its fMRI response using NRF; the predicted response is then fed into MindEye2, which reconstructs the corresponding visual stimulus. We compare these reconstructions against the ground-truth images presented during data collection.

PixCorr measures the pixel-wise correlation between the ground-truth image and the reconstruction. SSIM refers to the Structural Similarity Index, which evaluates perceptual similarity between ground-truth and reconstructed images. Alex(2), Alex(5), Incep, and CLIP are two-way identification metrics (chance = 50%) based on feature similarity. Specifically, Alex(2) uses features from the 2nd layer of AlexNet, Alex(5) from the 5th layer of AlexNet, Incep from the final pooling layer of InceptionV3, and CLIP from the final layer of CLIP ViT-L/14. In two-way identification, the task is to decide whether the voxel embedding is closer to its paired image embedding or to a randomly selected image embedding, reported as percent correct. Eff and SwAV denote representational similarity metrics, computed as the average correlation distance between voxel embeddings and features extracted from EfficientNet-B1 and SwAV-ResNet50, respectively.

**Interpretation of Semantic-Level Metrics**  The semantic-level reconstruction metrics reported in the main paper (PixCorr, SSIM, AlexNet features, Inception, CLIP, EfficientNet, SwAV) are computed using the MindEye2 fMRI-to-image decoder, which reconstructs visual stimuli from predicted fMRI responses. We included these semantic metrics to provide additional qualitative insight into the behavior of the generated or selected images, following the evaluation format adopted in recent prior work, including MindSimulator (Bao et al., 2025), published at ICLR 2025. Our intention was to maintain consistency with existing evaluation practices, rather than to treat these metrics as decisive evidence for model comparison. While useful for assessing whether predicted responses support plausible reconstructions, these metrics therefore require careful interpretation.

Because MindEye2 is itself a trained neural decoder, it introduces its own representational biases. As a result, we observe counterintuitive cases in which **fWRF predictions score higher than the actual measured fMRI responses** on some semantic reconstruction metrics. This does not imply that fWRF predictions are more neuronally accurate; rather, it reflects that the decoder's internal feature space is more compatible with certain statistical properties of the predicted responses than with the true fMRI signals.

This behavior demonstrates that semantic-level decoding metrics are not suitable as primary measures of encoding quality. Instead, they should be viewed as *diagnostic tools* for assessing whether predicted responses can support plausible image reconstructions.

Accordingly, all core model comparisons and statistical conclusions in the paper rely exclusively on voxel-level encoding metrics (e.g., Pearson correlation), which directly measure correspondence to ground-truth neural responses and are unaffected by decoder bias. Semantic reconstruction results are included for completeness and visualization, but should not be interpreted as primary indicators of model performance.

### A.4 ADDITIONAL RESULTS ON SINGLE SUBJECT NRF

Here we report the single subject NRF evaluation for all subjects (Subject 1, 2, 5, 7) in table 4 and the Mean ± SEM across 4 subjects in table 5.

| Subject | Method | Voxel-Level | | Semantic-Level (via decoding) | | | | | | | |
|---|---|---|---|---|---|---|---|---|---|---|---|
| | | Pearson↑ | MSE↓ | PixCorr↑ | SSIM↑ | Alex(2)↑ | Alex(5)↑ | IncepT↑ | CLIP↑ | Eff↓ | SwAV↓ |
| **Subject 1** | Linear Regression | 0.311 | 0.364 | 0.180 | 0.243 | 86.0% | 94.3% | 89.8% | 84.2% | 0.759 | 0.428 |
| | fWRF | 0.341 | 0.363 | 0.304 | 0.342 | 96.8% | 99.0% | 96.2% | 92.0% | 0.615 | 0.357 |
| | NRF (ours) | **0.361** | **0.349** | **0.324** | **0.387** | 95.6% | 98.3% | 94.1% | 91.9% | **0.680** | **0.396** |
| **Subject 2** | Linear Regression | 0.323 | 0.358 | 0.164 | 0.253 | 85.2% | 94.2% | 89.5% | 83.9% | 0.764 | 0.424 |
| | fWRF | 0.347 | 0.364 | 0.219 | 0.222 | 96.9% | 98.9% | 95.3% | 91.1% | 0.635 | 0.359 |
| | NRF (ours) | **0.368** | **0.347** | 0.240 | **0.351** | 89.3% | 95.4% | 88.1% | 87.5% | 0.767 | 0.442 |
| **Subject 5** | Linear Regression | 0.378 | 0.340 | 0.203 | 0.302 | 87.0% | 97.6% | 91.0% | 85.2% | 0.734 | 0.405 |
| | fWRF | 0.413 | 0.357 | 0.345 | 0.403 | 96.7% | 99.1% | 96.9% | 92.2% | 0.599 | 0.352 |
| | NRF (ours) | **0.425** | **0.346** | 0.236 | 0.379 | 93.5% | 97.9% | **97.1%** | **93.1%** | 0.609 | **0.317** |
| **Subject 7** | Linear Regression | 0.267 | 0.348 | 0.197 | 0.285 | 86.3% | 94.1% | 90.6% | 84.7% | 0.745 | 0.412 |
| | fWRF | 0.269 | 0.360 | 0.346 | 0.399 | 97.4% | 99.3% | 96.3% | 92.2% | 0.609 | 0.357 |
| | NRF (ours) | **0.278** | **0.328** | 0.244 | 0.367 | 88.0% | 93.6% | 89.2% | 84.6% | 0.768 | 0.445 |

Table 4: Single-subject NRF results for S1, S2, S5, and S7. For voxel-level metrics, the median values are reported.

| Method | Pearson↑ | MSE↓ | PixCorr↑ | SSIM↑ | Alex(2)↑ | Alex(5)↑ | IncepT↑ | CLIP↑ | Eff↓ | SwAV↓ |
|---|---|---|---|---|---|---|---|---|---|---|
| Linear Regression | 0.323 ± 0.021 | 0.353 ± 0.005 | 0.186 ± 0.009 | 0.271 ± 0.014 | 86.1%±0.4% | 95.0%±0.9% | 90.2%±0.3% | 84.5%±0.3% | 0.750 ± 0.007 | 0.417 ± 0.005 |
| fWRF | 0.343 ± 0.029 | 0.361 ± 0.002 | 0.303 ± 0.030 | 0.341 ± 0.042 | 96.9%±0.2% | 99.1%±0.1% | 96.2%±0.3% | 91.9%±0.3% | 0.614 ± 0.008 | 0.356 ± 0.001 |
| **NRF (ours)** | **0.358** ± 0.030 | **0.343** ± 0.005 | 0.261 ± 0.021 | 0.371 ± 0.008 | 91.6%±1.8% | 96.3%±1.1% | 92.1%±2.1% | 89.3%±3.6% | **0.706** ± 0.038 | **0.400** ± 0.030 |

Table 5: Mean ± SEM across all subjects. For voxel-level metrics, the mean of the per-subject medians is reported.

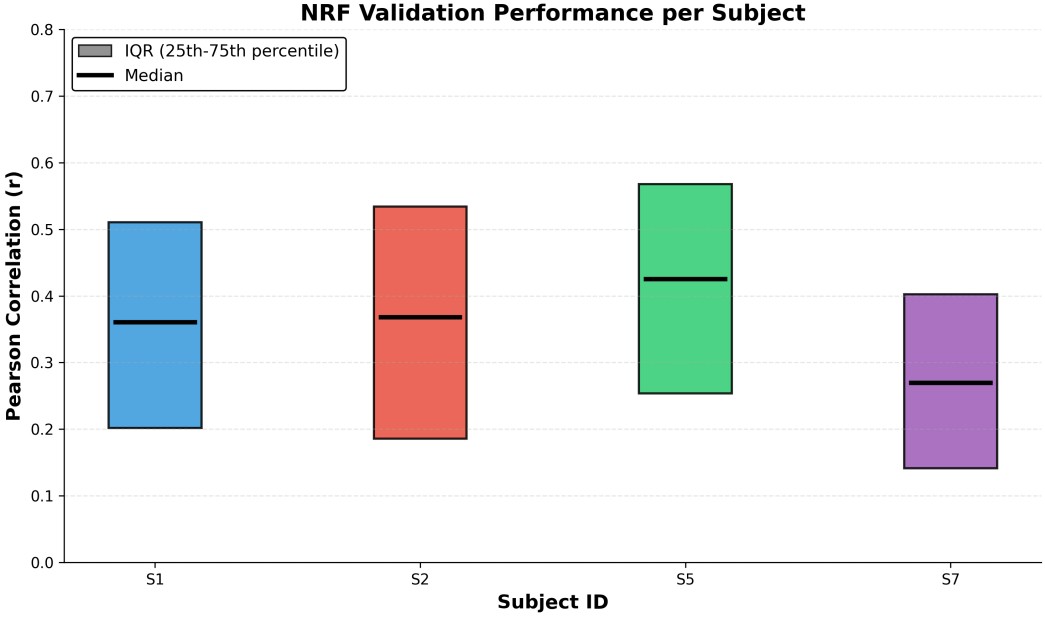

Figure 5: The distribution of the predicted voxel accuracy for 25th - 75th percentile for each subject

A.5    STAT SIGNIFICANCE ANALYSIS

This section provides the statistical procedures used to evaluate voxel-wise prediction accuracy, compare models, and analyze robustness across training-set sizes and ROI ablations. All analyses were repeated across multiple training sizes (20–800 images and full data; Fig. 2). For the ROI ablation experiments (Fig. 4), we applied the same paired permutation testing framework to compare performance between the original and spatially perturbed coordinate conditions.

**voxel-wise model comparison via paired permutation tests.**    To determine whether NRF significantly outperforms the baselines, we perform voxel-wise paired permutation tests between all model pairs (NRF vs. fWRF and NRF vs. Linear). For each voxel, we compute the correlation difference and evaluate its significance by constructing a null distribution from $10,000$ sign-flip permutations. Two-tailed p-values are corrected using the Benjamini–Hochberg FDR procedure ($\alpha = 0.05$), and effect sizes are reported using Cohen's $d$. Figure 2a reports the subject-level median correlations, while the full statistical results are provided in the following sections. Across all subjects, NRF shows significantly higher voxel-wise correlations than both baselines ($p_{\text{FDR}} < 0.05$), with medium-to-large effect sizes.

For the coordinate ablation experiments in Fig. 4, we apply the same paired permutation test to compare NRF performance under the original versus shuffled or shifted ROI coordinate conditions, allowing us to assess whether performance depends on anatomically aligned spatial inputs rather than subject-specific voxel patterns.

**Voxel-level predictive significance testing.**    We also assess whether each model produces statistically reliable voxel-wise predictions. For each subject and model, Pearson correlations between predicted and ground-truth responses are tested using the correlation t-test, followed by BH-FDR correction ($\alpha = 0.05$). NRF consistently yields a larger fraction of significant voxels than both baselines across all subjects, and this pattern holds across all training sizes.

For the ROI ablation experiments (Fig. 4), we perform the same voxel-level significance test to quantify how many voxels remain significant when coordinates are shuffled or spatially shifted. This allows us to evaluate whether anatomical misalignment degrades the reliability of voxel-wise predictions within targeted functional regions.

### A.5.1 SINGLE SUBJECT MODEL STATISTICS ANALYSIS (FIG 2A)

**Paired permutation test**

| Training images | Subject | $d$ (NRF vs fWRF) | $p_{FDR}$ | Sig? | $d$ (NRF vs Linear) | $p_{FDR}$ | Sig? |
|---|---|---|---|---|---|---|---|
| 20 | S1 | 0.40 | **0.000** | True | 0.48 | **0.000** | True |
| 20 | S2 | -0.01 | 0.126 | False | 0.37 | **0.000** | True |
| 20 | S5 | -0.22 | **0.000** | True | 0.14 | **0.000** | True |
| 20 | S7 | 0.06 | **0.000** | True | 0.24 | **0.000** | True |
| 60 | S1 | 0.21 | **0.000** | True | 0.71 | **0.000** | True |
| 60 | S2 | -0.12 | **0.000** | True | 0.50 | **0.000** | True |
| 60 | S5 | 0.08 | **0.000** | True | 0.59 | **0.000** | True |
| 60 | S7 | 0.06 | **0.000** | True | 0.57 | **0.000** | True |
| 200 | S1 | 0.36 | **0.000** | True | 0.74 | **0.000** | True |
| 200 | S2 | 0.20 | **0.000** | True | 0.76 | **0.000** | True |
| 200 | S5 | -0.06 | **0.000** | True | 0.55 | **0.000** | True |
| 200 | S7 | 0.20 | **0.000** | True | 0.50 | **0.000** | True |
| 400 | S1 | 0.32 | **0.000** | True | 0.68 | **0.000** | True |
| 400 | S2 | 0.30 | **0.000** | True | 0.65 | **0.000** | True |
| 400 | S5 | 0.15 | **0.000** | True | 0.55 | **0.000** | True |
| 400 | S7 | 0.20 | **0.000** | True | 0.51 | **0.000** | True |
| 600 | S1 | 0.26 | **0.000** | True | 0.62 | **0.000** | True |
| 600 | S2 | 0.39 | **0.000** | True | 0.64 | **0.000** | True |
| 600 | S5 | 0.23 | **0.000** | True | 0.51 | **0.000** | True |
| 600 | S7 | 0.08 | **0.000** | True | 0.40 | **0.000** | True |
| 800 | S1 | 0.47 | **0.000** | True | 0.61 | **0.000** | True |
| 800 | S2 | 0.37 | **0.000** | True | 0.60 | **0.000** | True |
| 800 | S5 | 0.17 | **0.000** | True | 0.43 | **0.000** | True |
| 800 | S7 | 0.03 | **0.0002** | True | 0.29 | **0.000** | True |
| Full | S1 | 0.30 | **0.000** | True | 0.50 | **0.000** | True |
| Full | S2 | 0.23 | **0.000** | True | 0.50 | **0.000** | True |
| Full | S5 | -0.08 | **0.000** | True | 0.08 | **0.000** | True |
| Full | S7 | 0.25 | **0.000** | True | 0.21 | **0.000** | True |

Table 6: Effect sizes (Cohen's $d$), BH–FDR corrected $p$-values, and significance for NRF vs. fWRF and NRF vs. Linear. For readability, $p$-values $< 0.0001$ are shown as 0.000.

**Aggregated results**

| Training images | $d$ (NRF vs fWRF) | $p_{FDR}$ NRF vs fWRF | # Sig | $d$ (NRF vs Linear) | $p_{FDR}$ (NRF vs Linear) | # Sig |
|---|---|---|---|---|---|---|
| 20 | $0.058 \pm 0.223$ | [0.000, 0.126] | 3/4 | $0.308 \pm 0.129$ | [0.000, 0.000] | 4/4 |
| 60 | $0.058 \pm 0.118$ | [0.000, 0.000] | 4/4 | $0.593 \pm 0.075$ | [0.000, 0.000] | 4/4 |
| 200 | $0.175 \pm 0.151$ | [0.000, 0.000] | 4/4 | $0.638 \pm 0.114$ | [0.000, 0.000] | 4/4 |
| 400 | $0.243 \pm 0.070$ | [0.000, 0.000] | 4/4 | $0.598 \pm 0.070$ | [0.000, 0.000] | 4/4 |
| 600 | $0.240 \pm 0.110$ | [0.000, 0.000] | 4/4 | $0.543 \pm 0.096$ | [0.000, 0.000] | 4/4 |
| 800 | $0.260 \pm 0.171$ | [0.000, 0.0002] | 4/4 | $0.483 \pm 0.132$ | [0.000, 0.000] | 4/4 |
| Full | $0.175 \pm 0.149$ | [0.000, 0.000] | 4/4 | $0.323 \pm 0.183$ | [0.000, 0.000] | 4/4 |

Table 7: Cross-subject mean $\pm$ SD effect sizes (Cohen's $d$), BH–FDR corrected $p$-value *ranges* across subjects, and the number of subjects with significant differences ($p < 0.05$). For readability, $p$-values $< 0.0001$ are shown as 0.000.

**Voxel-level significance testing**

| Training images | Subject | NRF (% sig) | fWRF (% sig) | Linear (% sig) | # Voxels |
|---|---|---|---|---|---|
| 20 | S1 | 67.20% | 51.93% | 48.32% | 15,724 |
| 20 | S2 | 67.63% | 62.91% | 49.37% | 14,278 |
| 20 | S5 | 69.23% | 69.65% | 60.94% | 13,039 |
| 20 | S7 | 52.38% | 50.38% | 38.82% | 12,682 |
| 60 | S1 | 80.60% | 73.40% | 62.62% | 15,724 |
| 60 | S2 | 75.83% | 69.97% | 57.91% | 14,278 |
| 60 | S5 | 83.89% | 79.15% | 69.45% | 13,039 |
| 60 | S7 | 70.23% | 67.32% | 53.22% | 12,682 |
| 200 | S1 | 86.31% | 81.47% | 71.97% | 15,724 |
| 200 | S2 | 85.00% | 80.21% | 72.71% | 14,278 |
| 200 | S5 | 89.32% | 88.26% | 82.82% | 13,039 |
| 200 | S7 | 77.69% | 73.91% | 69.35% | 12,682 |
| 400 | S1 | 89.12% | 87.38% | 79.78% | 15,724 |
| 400 | S2 | 87.23% | 84.71% | 80.76% | 14,278 |
| 400 | S5 | 91.61% | 90.77% | 88.35% | 13,039 |
| 400 | S7 | 84.32% | 81.08% | 76.72% | 12,682 |
| 600 | S1 | 89.91% | 88.69% | 82.94% | 15,724 |
| 600 | S2 | 88.76% | 86.47% | 83.95% | 14,278 |
| 600 | S5 | 92.64% | 91.71% | 91.01% | 13,039 |
| 600 | S7 | 82.10% | 82.58% | 79.08% | 12,682 |
| 800 | S1 | 90.68% | 88.44% | 84.75% | 15,724 |
| 800 | S2 | 89.01% | 86.72% | 84.65% | 14,278 |
| 800 | S5 | 93.13% | 92.87% | 91.84% | 13,039 |
| 800 | S7 | 84.57% | 84.18% | 83.14% | 12,682 |
| Full | S1 | 94.29% | 93.13% | 92.97% | 15,724 |
| Full | S2 | 92.23% | 91.06% | 90.95% | 14,278 |
| Full | S5 | 95.70% | 94.60% | 95.97% | 13,039 |
| Full | S7 | 89.90% | 88.63% | 90.18% | 12,682 |

Table 8: Fraction of voxels with significant prediction correlation (FDR-corrected) across training set sizes and subjects.

**Aggregated results**

| Training Images | NRF (%) | fWRF (%) | Linear (%) |
|---|---|---|---|
| 20 | **64.61 ± 7.22** | 58.22 ± 8.70 | 49.86 ± 8.98 |
| 60 | **77.14 ± 5.61** | 72.46 ± 5.41 | 60.80 ± 7.23 |
| 200 | **84.58 ± 4.80** | 81.21 ± 5.58 | 74.71 ± 5.48 |
| 400 | **88.57 ± 3.20** | 85.49 ± 3.92 | 81.40 ± 4.47 |
| 600 | **88.85 ± 4.20** | 87.36 ± 3.94 | 84.75 ± 5.60 |
| 800 | **89.85 ± 4.02** | 88.55 ± 3.92 | 86.09 ± 3.91 |
| Full | **93.03 ± 2.48** | 91.36 ± 2.09 | 92.52 ± 2.39 |

Table 9: Mean ± SD percentage of significant voxels across subjects for each training size.

### A.5.2 SHUFFLE COORDINATE ABLATION STATISTICS ANALYSIS (FIG 4A)

**Paired permutation test**

| Training Images | $d$ (Original vs Local) | $p_{\text{FDR}}$ | Sig? | $d$ (Original vs Global) | $p_{\text{FDR}}$ | Sig? |
|---|---|---|---|---|---|---|
| 20 | 0.202 | 0.000 | True | 0.816 | 0.000 | True |
| 100 | 0.422 | 0.000 | True | 0.831 | 0.000 | True |
| 200 | 0.877 | 0.000 | True | 1.024 | 0.000 | True |
| 400 | 0.780 | 0.000 | True | 0.956 | 0.000 | True |
| 600 | 0.737 | 0.000 | True | 0.985 | 0.000 | True |
| 800 | 0.773 | 0.000 | True | 0.892 | 0.000 | True |
| Full | 0.771 | 0.000 | True | 1.386 | 0.000 | True |

Table 10: Effect sizes (Cohen's $d$), BH–FDR corrected $p$-values, and significance for Original vs Coord shuffle Ablations. Original refers to original setting, while Local refer to ROI-wise shuffle and Global refer to Whole-Brain shuffle. For readability, $p$-values $< 0.0001$ are shown as 0.000.

**Voxel-level significance testing**

| Training images | Original (% sig) | Local (% sig) | Global (% sig) | # Voxels |
|---|---|---|---|---|
| 20 | 71.25% | 57.59% | 30.47% | 15001 |
| 100 | 86.75% | 80.17% | 65.11% | 15001 |
| 200 | 91.94% | 80.67% | 74.85% | 15001 |
| 400 | 92.61% | 85.21% | 80.99% | 15001 |
| 600 | 93.18% | 87.37% | 83.03% | 15001 |
| 800 | 93.66% | 88.11% | 86.73% | 15001 |
| Full | 95.31% | 91.97% | 86.17% | 15001 |

Table 11: Percentage of significant voxels for different settings. Original refers to original setting, while Local refer to ROI-wise shuffle and Global refer to Whole-Brain shuffle. For readability, $p$-values $< 0.0001$ are shown as 0.000.

### A.5.3 SHIFT COORDINATE ABLATION STATISTICS ANALYSIS (FIG 4B, C)

**Paired permutation test**

| Training images | $d$(Original EBA vs Shifted EBA) | $p_{\text{FDR}}$ | Sig? |
|:---:|:---:|:---:|:---|
| 20 | 0.627 | 0.000 | True |
| 100 | 0.128 | 0.000 | True |
| 200 | −0.114 | 0.000 | True |
| 400 | 0.276 | 0.000 | True |
| 600 | 0.244 | 0.000 | True |
| 800 | 0.115 | 0.000 | True |

Table 12: Effect sizes (Cohen's $d$), BH–FDR corrected $p$-values, and significance for Original vs Coord shift ablation. Original refers to original setting, while shifted correspond to finetuning on shifted EBA voxels. For readability, $p$-values $< 0.0001$ are shown as 0.000.

| Training images | $d$(Original V1 vs Shifted V1) | $p_{\text{FDR}}$ | Sig? |
|:---:|:---:|:---:|:---|
| 20 | 0.609 | 0.000 | True |
| 100 | 0.550 | 0.000 | True |
| 200 | 0.415 | 0.000 | True |
| 400 | 0.204 | 0.000 | True |
| 600 | 0.299 | 0.000 | True |
| 800 | 0.302 | 0.000 | True |

Table 13: Effect sizes (Cohen's $d$), BH–FDR corrected $p$-values, and significance for Original vs Coord shift ablation. Original refers to original setting, while shifted correspond to finetuning on shifted V1 voxels. For readability, $p$-values $< 0.0001$ are shown as 0.000.

**Voxel-level significance testing**

| Training images | Original_EBA (% sig) | Shifted_EBA (% sig) | # Voxels |
|:---:|:---:|:---:|:---:|
| 20 | 69.16% | 50.50% | 3184 |
| 100 | 75.88% | 69.91% | 3184 |
| 200 | 73.08% | 73.71% | 3184 |
| 400 | 79.55% | 76.73% | 3184 |
| 600 | 78.33% | 76.35% | 3184 |
| 800 | 78.80% | 77.10% | 3184 |

Table 14: Percentage of significant EBA voxels for Original EBA and Shifted EBA.

| Training images | Original V1 (% sig) | Shifted V1 (% sig) | # Voxels |
|:---:|:---:|:---:|:---:|
| 20 | 73.93% | 63.78% | 1074 |
| 100 | 85.94% | 84.64% | 1074 |
| 200 | 85.66% | 83.43% | 1074 |
| 400 | 89.11% | 86.78% | 1074 |
| 600 | 88.27% | 86.22% | 1074 |
| 800 | 87.24% | 85.75% | 1074 |

Table 15: Percentage of significant V1 voxels for Original V1 and Shifted V1.

## A.6 ADDITIONAL SUBJECT ADAPTATION RESULT

Here, we present additional results of new subject adaptation for subjects 1, 2, 5 and 7 in Table 16, Table 17, Table 18, and Table 19 respectively. The results show that our method consistently yields superior performance compared to the scratch method. We show the and the mean ± SEM across the four subjects in Table 20.

| Training Images | Method | Voxel-Level | | Semantic-Level (via decoding) | | | | | | | |
|---|---|---|---|---|---|---|---|---|---|---|---|
| | | Pearson↑ | MSE↓ | PixCorr↑ | SSIM↑ | Alex(2)↑ | Alex(5)↑ | IncepT↑ | CLIP↑ | Eff↓ | SwAV↓ |
| 20 | NRF scratch | 0.116 | **0.411** | 0.023 | 0.163 | 0.548 | 0.552 | 56.8% | 53.9% | 0.969 | 0.660 |
| | NRF finetune ensemble | **0.184** | 0.463 | **0.139** | **0.308** | **0.744** | **0.809** | **72.5%** | **68.9%** | **0.885** | **0.540** |
| 200 | NRF scratch | 0.261 | **0.377** | 0.132 | 0.242 | 0.750 | 0.811 | 73.6% | 70.3% | 0.892 | 0.555 |
| | NRF finetune ensemble | **0.306** | 0.379 | **0.266** | **0.375** | **0.917** | **0.958** | **88.2%** | **85.9%** | **0.758** | **0.437** |
| 800 | NRF scratch | 0.314 | 0.369 | 0.244 | 0.307 | 0.915 | 0.962 | 90.6% | 86.1% | 0.742 | 0.432 |
| | NRF finetune ensemble | **0.342** | **0.361** | **0.316** | **0.382** | **0.945** | **0.980** | **93.3%** | **88.7%** | **0.698** | **0.404** |

Table 16: New subject adaptation with limited data (20, 200, 800 images). NRF pretrained on subjects 2,5,7 are used as base models to adapt to subject 1.

| Training Images | Method | Voxel-Level | | Semantic-Level (via decoding) | | | | | | | |
|---|---|---|---|---|---|---|---|---|---|---|---|
| | | Pearson↑ | MSE↓ | PixCorr↑ | SSIM↑ | Alex(2)↑ | Alex(5)↑ | IncepT↑ | CLIP↑ | Eff↓ | SwAV↓ |
| 20 | NRF scratch | 0.124 | **0.462** | 0.023 | **0.344** | 0.499 | 0.498 | 49.6% | 49.6% | 0.971 | 0.636 |
| | NRF finetune ensemble | **0.168** | 0.457 | **0.140** | 0.328 | **0.780** | **0.848** | **74.0%** | **67.9%** | **0.874** | **0.532** |
| 200 | NRF scratch | 0.266 | **0.386** | 0.076 | 0.323 | 0.617 | 0.674 | 62.7% | 57.8% | 0.944 | 0.605 |
| | NRF finetune ensemble | **0.317** | 0.375 | **0.255** | **0.365** | **0.916** | **0.966** | **90.3%** | **85.5%** | **0.735** | **0.427** |
| 800 | NRF scratch | 0.323 | 0.372 | 0.192 | 0.299 | 0.885 | 0.951 | 87.4% | 82.5% | 0.778 | 0.452 |
| | NRF finetune ensemble | **0.356** | **0.354** | **0.274** | **0.374** | **0.935** | **0.976** | **92.4%** | **88.0%** | **0.711** | **0.413** |

Table 17: New subject adaptation with limited data (20, 200, 800 images). NRF pretrained on subjects 1,5,7 are used as base models to adapt to subject 2.

| Training Images | Method | Voxel-Level | | Semantic-Level (via decoding) | | | | | | | |
|---|---|---|---|---|---|---|---|---|---|---|---|
| | | Pearson↑ | MSE↓ | PixCorr↑ | SSIM↑ | Alex(2)↑ | Alex(5)↑ | IncepT↑ | CLIP↑ | Eff↓ | SwAV↓ |
| 20 | NRF scratch | 0.128 | **0.433** | 0.126 | 0.213 | **0.608** | **0.633** | 56.6% | 56.8% | 0.953 | 0.592 |
| | NRF finetune ensemble | **0.184** | 0.470 | **0.161** | **0.366** | 0.695 | 0.744 | **70.2%** | **67.3%** | **0.889** | **0.533** |
| 200 | NRF scratch | 0.293 | **0.415** | 0.113 | 0.238 | 0.687 | 0.719 | 68.0% | 64.3% | 0.911 | 0.573 |
| | NRF finetune ensemble | **0.353** | 0.373 | **0.242** | **0.374** | **0.882** | **0.936** | **88.4%** | **84.8%** | **0.765** | **0.445** |
| 800 | NRF scratch | 0.365 | 0.370 | 0.211 | 0.326 | 0.883 | 0.945 | 89.3% | 87.0% | 0.734 | 0.426 |
| | NRF finetune ensemble | **0.400** | **0.355** | **0.259** | **0.383** | **0.916** | **0.969** | **92.8%** | **90.9%** | **0.682** | **0.401** |

Table 18: New subject adaptation with limited data (20, 200, 800 images). NRF pretrained on subjects 1,2,7 are used as base models to adapt to subject 5.

| Training Images | Method | Voxel-Level | | Semantic-Level (via decoding) | | | | | | | |
|---|---|---|---|---|---|---|---|---|---|---|---|
| | | Pearson↑ | MSE↓ | PixCorr↑ | SSIM↑ | Alex(2)↑ | Alex(5)↑ | IncepT↑ (%) | CLIP↑ (%) | Eff↓ | SwAV↓ |
| Full | NRF subject 7 (all data) | 0.269 | 0.348 | 0.244 | 0.367 | 0.880 | 0.936 | 89.2% | 84.6% | 0.768 | 0.445 |
| 20 | NRF scratch | 0.076 | **0.417** | 0.060 | 0.195 | 0.564 | 0.597 | 54.9% | 54.5% | 0.962 | 0.621 |
| | NRF finetune ensemble | **0.114** | 0.445 | **0.186** | **0.366** | **0.750** | **0.792** | **73.2%** | **72.9%** | **0.868** | **0.515** |
| 200 | NRF scratch | 0.180 | 0.394 | 0.159 | 0.284 | 0.760 | 0.813 | 77.4% | 71.6% | 0.857 | 0.515 |
| | NRF finetune ensemble | **0.227** | **0.390** | **0.255** | **0.372** | **0.908** | **0.957** | **91.3%** | **87.3%** | **0.729** | **0.425** |
| 800 | NRF scratch | 0.220 | 0.376 | 0.188 | 0.313 | 0.856 | 0.926 | 87.8% | 83.4% | 0.772 | 0.452 |
| | NRF finetune ensemble | **0.251** | **0.372** | **0.269** | **0.382** | **0.927** | **0.970** | **92.2%** | **89.5%** | **0.700** | **0.408** |

Table 19: New subject adaptation with limited data (20, 200, 800 images). NRF pretrained on subjects 1,2,5 are used as base models to adapt to subject 7.

| Training Images | Method | Voxel-Level | | Semantic-Level (via decoding) | | | | | | | |
|---|---|---|---|---|---|---|---|---|---|---|---|
| | | Pearson↑ | MSE↓ | PixCorr↑ | SSIM↑ | Alex(2)↑ | Alex(5)↑ | IncepT↑ (%) | CLIP↑ (%) | Eff↓ | SwAV↓ |
| 20 | NRF scratch | 0.111 ± 0.012 | 0.431 ± 0.011 | 0.058 ± 0.024 | 0.229 ± 0.040 | 0.555 ± 0.023 | 0.570 ± 0.029 | 54.5 ± 1.7 | 53.7 ± 1.5 | 0.964 ± 0.004 | 0.627 ± 0.014 |
| | NRF finetune ensemble | 0.163 ± 0.017 | 0.459 ± 0.005 | 0.157 ± 0.011 | 0.342 ± 0.014 | 0.742 ± 0.018 | 0.798 ± 0.022 | 72.5 ± 0.8 | 69.3 ± 1.3 | 0.879 ± 0.005 | 0.530 ± 0.005 |
| 200 | NRF scratch | 0.250 ± 0.024 | 0.393 ± 0.008 | 0.120 ± 0.017 | 0.272 ± 0.020 | 0.704 ± 0.033 | 0.754 ± 0.035 | 70.4 ± 3.2 | 66.0 ± 3.2 | 0.901 ± 0.018 | 0.562 ± 0.019 |
| | NRF finetune ensemble | 0.301 ± 0.027 | 0.379 ± 0.004 | 0.255 ± 0.005 | 0.372 ± 0.002 | 0.906 ± 0.008 | 0.954 ± 0.006 | 89.6 ± 0.8 | 85.9 ± 0.5 | 0.747 ± 0.009 | 0.434 ± 0.005 |
| 800 | NRF scratch | 0.306 ± 0.031 | 0.372 ± 0.002 | 0.209 ± 0.013 | 0.311 ± 0.006 | 0.885 ± 0.012 | 0.946 ± 0.008 | 88.8 ± 0.7 | 84.8 ± 1.1 | 0.757 ± 0.011 | 0.441 ± 0.007 |
| | NRF finetune ensemble | 0.337 ± 0.031 | 0.360 ± 0.004 | 0.280 ± 0.013 | 0.380 ± 0.002 | 0.931 ± 0.006 | 0.974 ± 0.003 | 92.7 ± 0.2 | 89.3 ± 0.6 | 0.698 ± 0.006 | 0.407 ± 0.003 |

Table 20: Mean ± SEM across subjects (S1, S2, S5, S7) for all training sizes and methods.

## A.7 Additional ablation results

**Finetuning strategy**  During finetuning, there are two options: the projection model and the image feature merger. We explored finetuning both/projector-only and the feature extraction block only. We finetuned subject 1 NRF with 800 images from subject 8 data. The results are shown in the Table A.7. The result suggests that finetuning should be done on both components to get the maximum performance boost.

| Finetune strategy | Voxel-Level | | Semantic-Level (via decoding) | | | | | | | |
|---|---|---|---|---|---|---|---|---|---|---|
| | Pearson↑ | MSE↓ | PixCorr↑ | SSIM↑ | Alex(2)↑ | Alex(5)↑ | IncepT↑ | CLIP↑ | Eff↓ | SwAV↓ |
| Both | 0.234 | 0.361 | 0.254 | 0.377 | 0.910 | 0.958 | 90.9% | 86.7% | 0.730 | 0.424 |
| Image extractor only | 0.104 | 0.394 | 0.102 | 0.274 | 0.613 | 0.636 | 59.2% | 55.6% | 0.958 | 0.596 |
| Projector only | 0.233 | 0.364 | 0.249 | 0.361 | 0.894 | 0.945 | 88.8% | 83.8% | 0.766 | 0.446 |

Table 21: Ablation of different finetuning strategies. Models pre-trained on subject 1 finetuned on 800 images from subject 7.

**Number of layers**  To investigate the effect of model architecture on neural response prediction accuracy, we conducted an ablation study by varying the number of layers and the hidden dimension of the MLP projector. As shown in Table 22, we computed 4, 8, 16-layer configurations under subject-specific settings on subject1. The results show that 8-layer model results in the best performance, which is also the setting we utilized for our experiments.

| Layers | Voxel-Level | | Semantic-Level (via decoding) | | | | | | | |
|---|---|---|---|---|---|---|---|---|---|---|
| | Pearson↑ | MSE↓ | PixCorr↑ | SSIM↑ | Alex(2)↑ | Alex(5)↑ | IncepT↑ | CLIP↑ | Eff↓ | SwAV↓ |
| 4 | 0.358 | 0.348 | 0.258 | 0.351 | 0.914 | 0.961 | 89.7% | 85.1% | 0.748 | 0.437 |
| 8 | 0.360 | 0.350 | 0.324 | 0.387 | 0.956 | 0.983 | 94.1% | 89.9% | 0.680 | 0.396 |
| 16 | 0.349 | 0.353 | 0.331 | 0.385 | 0.956 | 0.983 | 93.99% | 89.9% | 0.678 | 0.395 |

Table 22: Ablation of the number of layers of the MLP projector. Model trained on subject 1 data.

**Hidden dimension**  We also conducted an ablation study by varying the hidden dimension of the MLP projector. As shown in Table, we computed for hidden dimension = 2048, 4096, 8192 configurations under subject-specific settings on subject1. The results show that model with hidden dimension = 4096 results in the best performance, which is also the setting we utilized for our experiments.

| Hidden dimension | Voxel-Level | | Semantic-Level (via decoding) | | | | | | | |
|---|---|---|---|---|---|---|---|---|---|---|
| | Pearson↑ | MSE↓ | PixCorr↑ | SSIM↑ | Alex(2)↑ | Alex(5)↑ | IncepT↑ | CLIP↑ | Eff↓ | SwAV↓ |
| 2048 | 0.358 | 0.348 | 0.253 | 0.353 | 0.903 | 0.948 | 87.4% | 83.2% | 0.777 | 0.451 |
| 4096 | 0.360 | 0.350 | 0.324 | 0.387 | 0.956 | 0.983 | 94.1% | 89.9% | 0.680 | 0.396 |
| 8192 | 0.353 | 0.353 | 0.323 | 0.386 | 0.955 | 0.981 | 94.1% | 89.9% | 0.683 | 0.396 |

Table 23: Ablation on hidden layer dimension of the MLP projector. Model trained on subject 1 data.

## A.8 HIGH-RESOLUTION NRF QUERYING

Once trained, NRF behaves as a continuous "digital twin" of the voxel response field: given an image embedding and any 3D anatomical coordinate, NRF directly outputs the predicted response at that location. Unlike traditional voxel-wise models, NRF does *not* depend on the discrete voxel grid of the acquired fMRI data.

Because NRF learns a continuous neural field over standardized brain space, it can be queried on spatial grids of arbitrary resolution e.g., 0.5 mm or finer—without any volumetric resampling, interpolation, or reconstruction of the fMRI volume. This completely bypasses the computationally expensive and resolution-dependent resampling pipeline typically required for aligning or upsampling fMRI data.

Figure 6 illustrates this capability by comparing NRF predictions evaluated on a 0.5 mm grid with the original measured responses acquired at 1.8 mm resolution.

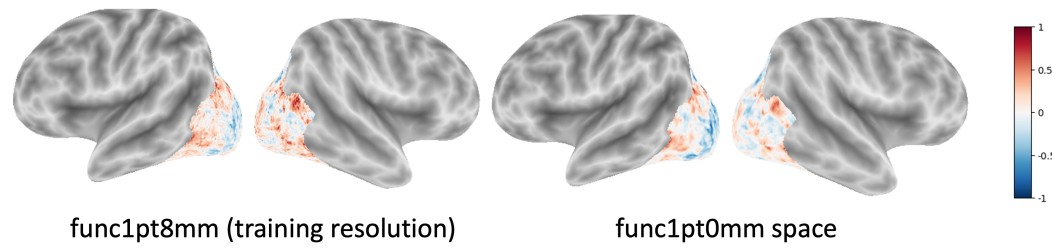

Figure 6: **NRF enables higher-resolution querying of the learned response field.** Because NRF models brain activity as a continuous function over 3D MNI space, it can be queried at arbitrary spatial locations after training. Shown here is an example slice from Subject 1: *left*, NRF predictions at the original measured func1pt8mm space *right*, the NRF predictions on a higher-resolution func1pt0mm space. NRF's continuous formulation enables principled, high-resolution resampling of the predicted response field beyond the acquired voxel grid.

# B    USE OF LLMS

LLMs were used only for polishing grammar and writing clarity.

# C    ETHIC STATEMENT

Our research adheres to the ICLR Code of Ethics. All experiments in this paper are conducted using open-source datasets, and no potential ethical concerns are associated with this work.

# D    REPRODUCIBILITY STATEMENT

All preprocessed data, code, and model parameters used in our research will be made publicly available upon publication. Detailed protocols for data preprocessing, model training, and evaluation have been provided in our manuscript, enabling independent reproduction.

