# OpenReview forum: "Beyond Grid-Locked Voxels: Neural Response Functions for Continuous Brain Encoding"
_ICLR.cc/2026/Conference — ICLR 2026 Poster_

### Official Review · Reviewer_vUPt · 2025-10-26

**Soundness:** 3
**Presentation:** 3
**Contribution:** 3
**Rating:** 2
**Confidence:** 4

**Summary:**

This paper adapts Neural Radiance Fields (NeRF) for the fMRI-to-image encoding task. In contrast to the common approach of predicting each voxel independently, this method predicts all voxel activations in a coupled manner. This strategy yields locally smooth predictions and facilitates efficient transfer learning.

**Strengths:**

The paper presents an original approach to the challenging task of predicting fMRI activations in a structured and meaningful way.
The framework also facilitates an interesting subject adaptation method.

**Weaknesses:**

- The reported encoding performance is very low. On the NSD dataset, a decent model for the selected voxels should yield correlations above 0.5 for subjects 1, 2, and 5, with strong models achieving values around 0.6. Notably, other works (e.g., [1, 2]) report significantly better performance on a wider selection of NSD voxels.

- Applying the fWRF approach to layers from DinoV2/CLIP, combined with LoRa adaptation and well-tuned hyperparameters, would outperform the reported results by a larger margin than the one shown between the NRF method and other baselines.

- The fWRF model outperforms the proposed approach on all but one reported decoding metrics.

References:

[1] Adeli, H., Sun, M., & Kriegeskorte, N. (2025). Transformer brain encoders explain human high-level visual responses. arXiv preprint arXiv:2505.17329.

[2] Beliy, R., Wasserman, N., Zalcher, A., & Irani, M. (2024). The Wisdom of a Crowd of Brains: A Universal Brain Encoder. arXiv preprint arXiv:2406.12179.

**Questions:**

- Can your approach be combined with competitive encoding models?
- Does your method enable an increase in the spatial resolution of the fMRI signal, or is the benefit of local smoothness limited to the predictions themselves?

---

> ### Author Response · Authors · 2025-11-23
> **Response 1/2 to Reviewer vUPt**
>
> We thank the reviewer for their very thoughtful comments and feedback! Please see below for our responses.
>
> >**W1:_Low reported encoding accuracy_**
>
> **Response:**  Thank you for raising this point. There are some misunderstandings regarding the encoding accuracy comparison.
>
> **1) Clarification about our reported metric**
> In the main paper, we reported **only the median voxelwise prediction accuracy across all modeled voxels**. This single summary statistic is conservative and naturally appears lower than works that report the **full distribution** (e.g., Q1–Q3). To provide a more complete picture, we have added the voxelwise prediction distribution plot(Q1–Q3) in the **Appendix A4** . Below we report the quartiles for each subject. These values show that NRF achieves **Q3 correlations above 0.5** for subjects S1, S2, and S5—consistent with the reviewer’s expectations and with strong prior models.
>
> | Statistic | S1      | S2      | S5      | S7      |
> |-----------|---------|---------|---------|---------|
> | Q1        | 0.2016  | 0.1858  | 0.2536  | 0.1412  |
> | Median    | 0.3605  | 0.3683  | 0.4253  | 0.2696  |
> | Q3        | **0.5104** | **0.5340** | **0.5676** | 0.4023 |
>
> **2) Clarifying differences in metrics used in prior work [1]**
> Paper [1] reports **explained variance**, defined as  **r^2 / noise ceiling**.
> This metric is not directly comparable to our reported Pearson correlation r. Following their practice, we computed explained variance for NRF and compared it to the models in [1] trained with a similar CLIP vision backbone. As shown below, **NRF achieves comparable explained variance and outperforms these models on several subjects**.
>
> | Model                  | S1    | S2    | S5    | S7    |
> |------------------------|-------|-------|-------|-------|
> | Transformer (ROIs)     | 0.53  | 0.49  | 0.50  | 0.38  |
> | Transformer (Vertices) | 0.55  | 0.52  | 0.52  | 0.40  |
> | **NRF**                | 0.50  | **0.52** | **0.55** | **0.42** |
>
> **3) Clarifying comparison to paper [2]**
> Paper [2] reports the **full voxelwise correlation distribution** (Q1–Q3), whereas our main paper reported only the **median**, which understates the performance of high-SNR voxels and makes direct comparison misleading. When evaluated using the same statistics—Q1, median, and Q3—NRF’s performance closely matches the distributions reported in [2], with Q3 values exceeding 0.5 for subjects S1, S2, and S5.
>
> **Summary:** The perceived performance discrepancy arises from **different reporting conventions**, not limitations of NRF. Under comparable metrics, NRF performs on par with strong baselines and prior work. We will cite these related encoding studies in the revised version.
>
>
> >**W2:_New Approaches_**
>
> **Response:** We thank the reviewer for this thoughtful suggestion. The direction proposed is indeed interesting and worth exploring. As clarified in **response to W1**, the perceived performance gap in our comparisons arises primarily from **differences in reporting conventions** (median-only vs. full voxelwise distributions), rather than from limitations of NRF. When evaluated using comparable metrics such as Q1–Q3 or explained variance, NRF performs on par with strong existing models.
>
> We agree that extending fWRF to modern vision backbones (e.g., DINOv2, CLIP) and incorporating parameter-efficient fine-tuning methods such as LoRA is a promising avenue for building even stronger baselines.  This direction is complementary to our approach, and we view it as a valuable opportunity for future research!
>
> >**W3:_Semantic metrics_**
>
> **Response:** We thank the reviewer for pointing this point out. In the “Response to All Reviewers,” we have included a discussion clarifying the interpretation of the semantic evaluation metric. We have also added a subsection in **Appendix A.3** that further analyzes this issue and clarifying the limitations of using such semantic metrics for comparing neural encoding models.

---

> > ### Comment · Reviewer_vUPt · 2025-11-24
> >
> > **W1:Low reported encoding accuracy**
> >
> > The main reported results in the paper [1] Table1 are (0.60 0.56 0.56 0.42 for sub S1,S2,S5,S7), not the results you are showing.
> > Paper [2] provides **median correlation** per subject on a subset of ~40K (this paper select s subset of ~15K voxels), for this subset the paper reports a median correlation above 0.5 for sub1, sub2, sub5 and above 0.4 for sub7. This model on your voxel selection would yield a much higher median correlation.
> >
> > **The encoding results fall considerably short of current standards.**
> >
> > The reason the model underperforms is because it doesn't take into account retinotopic structure and Population Receptive Fields.
> >
> > **W3:Semantic metrics**
> >
> > It is inconsistent to argue against the validity of metrics that were selected by the authors to report the paper's main results.

---

> > > ### Author Response · Authors · 2025-11-26
> > > **Clarification Regarding Encoding Accuracy Comparison**
> > >
> > > We thank the reviewer for their feedback. Please see below for our responses.
> > >
> > > >**W1:_Low reported encoding accuracy_**
> > >
> > > **Response:**
> > >
> > > ### **1) Regarding Paper [1]**
> > >
> > > We thank the reviewer for pointing this out. In our initial response, we compared against the performance of models using the **CLIP vision backbone**, consistent with what we employed.
> > >
> > > Upon further investigation of the referenced paper [1], we noted that their evaluation protocol specifies that:
> > >
> > > > **“For all models including all the baselines, we did 10-fold cross-validation using the training set for each subject and averaged the model predictions across all folds.”**
> > >
> > > Our previously reported comparison was based on predictions from a single training run. To ensure a fair and consistent comparison, we have now
> > > - Re-evaluated our model using the exact same 10-fold cross-validation protocol and averaged predictions across all folds
> > > - Report the explained variance of the averaged responses below.
> > >
> > > **Explained Variance Comparison (10-Fold Averaged)**
> > >
> > > | Model                        | S1     | S2     | S5     | S7     |
> > > |------------------------------|--------|--------|--------|--------|
> > > | CLIP Transformer (ROIs)      | 0.53   | 0.49   | 0.50   | 0.38   |
> > > | CLIP Transformer (Vertices) | 0.55   | 0.52   | 0.52   | 0.40   |
> > > | DINO Transformer (ROIs)     | 0.60   | 0.56   | 0.56   | 0.42   |
> > > | **NRF (Ours)**               | 0.5730 | **0.5734** | **0.5960** | **0.4789** |
> > >
> > > For full transparency, we additionally report the **per-fold explained variance across the 10 folds**:
> > >
> > > | Subject | Fold 1 | Fold 2 | Fold 3 | Fold 4 | Fold 5 | Fold 6 | Fold 7 | Fold 8 | Fold 9 | Fold 10 |
> > > |---------|--------|--------|--------|--------|--------|--------|--------|--------|--------|---------|
> > > | S1 | 0.5204 | 0.5117 | 0.5042 | 0.5047 | 0.5027 | 0.5105 | 0.5094 | 0.5168 | 0.5110 | 0.5134 |
> > > | S2 | 0.5101 | 0.4995 | 0.5224 | 0.5152 | 0.5226 | 0.5211 | 0.5185 | 0.5012 | 0.5099 | 0.5117 |
> > > | S5 | 0.5516 | 0.5477 | 0.5648 | 0.5607 | 0.5441 | 0.5383 | 0.5570 | 0.5371 | 0.5440 | 0.5721 |
> > > | S7 | 0.4347 | 0.4120 | 0.4441 | 0.4209 | 0.4438 | 0.4348 | 0.4428 | 0.4226 | 0.4324 | 0.4073 |
> > >
> > > From this revised comparison, we observe that **our model outperforms the CLIP-based results in [1] and further exceeds the DINO-based results (Table 1) on several subjects** under a strictly matched cross-validation protocol. More importantly, our method additionally demonstrates efficient cross-subject adaptation and enables resampling of fMRI responses on higher-resolution grids, which are not demonstrated by [1].
> > >
> > > ---
> > >
> > > ### **2) Regarding Paper [2]**
> > >
> > > We acknowledge that [2] reports median correlation values on a different and substantially larger voxel subset (~40K voxels), whereas our evaluation is conducted on a subset of ~15K voxels selected using standard visual ROI criteria. However the voxel selection, preprocessing, normalization, and denoising procedures in [2] are **not publicly released**, and the work itself is currently **unpublished with no available codebase**. As a result, we are unable to **perform a fair and controlled comparison under matched voxel selection and preprocessing conditions**.
> > >
> > > Importantly, reporting higher median correlations on a different voxel subset **does not necessarily imply superior model performance**, as voxel selection alone can significantly shift absolute correlation values depending on the signal-to-noise characteristics of the included voxels.
> > >
> > > ---
> > >
> > > ### **3) Regarding Retinotopy and Population Receptive Fields (pRFs)**
> > >
> > > We respectfully disagree with the statement that our model underperforms because it does not explicitly model retinotopic structure or population receptive fields. Notably, the transformer-based baselines in [1] also do not explicitly incorporate pRF modeling, yet achieve strong encoding performance. This suggests that **explicit pRF modeling is not required to obtain high prediction accuracy**.
> > >
> > > Instead, our method introduces spatial structure directly through continuous 3D anatomical coordinates, which encourages **smooth and spatially consistent predictions across the cortex**. Our results further demonstrate that this design choice yields performance that is competitive with other published state-of-the-art methods. This provides a different but equally valid way of incorporating spatial structure into the model.

---

> > > ### Author Response · Authors · 2025-11-26
> > > **Clarification Regarding Semantic metrics**
> > >
> > > > **W3:_Semantic metrics_**
> > >
> > > **Response:** We thank the reviewer for raising this point. While we do report semantic metrics in the paper, we would like to clarify that these metrics are used strictly as a diagnostic tool rather than as a primary comparison metric. Importantly, **none of the main conclusions in our paper are drawn from comparisons based on the semantic metrics**; all core claims are supported by **voxel-level encoding metrics**.
> > >
> > > We included the semantic metrics to provide additional qualitative insight into the behavior of the generated or selected images,  following the evaluation format adopted in recent prior work, including **MindSimulator [3], published at ICLR 2025**. Our intention was to maintain consistency with existing evaluation practices, rather than to treat these metrics as decisive evidence for model comparison.
> > >
> > > That said, to avoid any potential confusion, we are happy to **move the semantic metric results to the appendix** and retain only the primary voxel-level metrics in the main paper. We believe this preserves the clarity of our main contributions while still allowing interested readers to access the diagnostic analysis.

---

> > > ### Author Response · Authors · 2025-11-26
> > > **Reference**
> > >
> > > [1] Adeli, H., Sun, M., & Kriegeskorte, N. (2025). Transformer brain encoders explain human high-level visual responses. arXiv preprint arXiv:2505.17329.
> > >
> > > [2] Beliy, R., Wasserman, N., Zalcher, A., & Irani, M. (2024). The Wisdom of a Crowd of Brains: A Universal Brain Encoder. arXiv preprint arXiv:2406.12179.
> > >
> > > [3] Bao et al. (2025). MindSimulator: Exploring Brain Concept Localization via Synthetic fMRI. ICLR 2025. arXiv preprint at arXiv:2503.02351.

---

> ### Author Response · Authors · 2025-11-23
> **Response 2/2 to Reviewer vUPt**
>
> >**_Q1:Combining with competitive encoding models_**
>
> **Response:** We thank the reviewer for this question. NRF’s coordinate-aware formulation is fully compatible with transformer-based encoding architectures. The method is not tied to an MLP; it only requires that voxel coordinates (after positional encoding) be fused with image-derived features. This fusion can occur at the input layer, as an additional token, or through cross-attention within a transformer. Replacing the MLP in NRF with a transformer decoder or a cross-attention block is therefore straightforward and could enable richer interactions between spatial coordinates and visual features. While this exploration is beyond the scope of the current paper, the framework is inherently flexible and can naturally integrate with competitive transformer-based encoding models in future work.
>
>
>
> >**_Q2:Does NRF increase the spatial resolution of the fMRI signal, or is the smoothness limited to the predictions themselves?_**
>
> **Response:** We thank the reviewer for this insightful point. Once NRF is trained, it functions as a continuous “digital twin’’ of the voxel response field. Because NRF represents neural activity as a **continuous function over 3D anatomical space**, it can be queried at any coordinate—regardless of the resolution of the measured voxel grid.
> This enables inference on much finer spatial grids (e.g., 0.5 mm) without any volumetric resampling or interpolation of the fMRI data. In this sense, NRF provides a principled way to obtain higher-resolution predictions of the learned response field, beyond the discrete voxels that were actually acquired. We have added an example in the **Appendix A.8**, where we compare NRF predictions evaluated on a 0.5 mm grid with the original 1.8 mm measurements.
> We see this capability as a promising direction for **more flexible, resolution-agnostic fMRI data analysis**, enabling neuroscientists to inspect and visualize response fields at arbitrary spatial scales without relying on traditional resampling pipelines.

---

### Official Review · Reviewer_vXEp · 2025-10-28

**Soundness:** 3
**Presentation:** 4
**Contribution:** 3
**Rating:** 8
**Confidence:** 3

**Summary:**

The authors propose an implicit neural representation method for fMRI data, called neural response function (NRF). The proposed method utilizes local smoothness and cross-subject alignment to learn a function that can be used to predict the voxel-level response to a given stimulus continuously over R^3 space. This is presented as a significant improvement over the current state of the art methods that generally flatten 3d voxel space into a 1d vector and independently predict brain-response at each location. Which means current approaches break the anatomical reality where voxel level response is often spatially correlated and it also means cross-subject prediction is impossible as two voxels for two different subjects aren't comparable. NRF uses positional embeddings (with learnable coefficients) to embed voxel locations and then concatenates those with learnable image (stimulus) embedding and then passes these through a learnable NN model to predict the response for a subject's given voxel to the given image. Once pretrained on a single subject, the model can then be finetuned on partial data from other subjects to predict that subject's brain response. Multiple such pretrained models can be combined using regression and finetuned on a new subject to improve this prediction even further.

**Strengths:**

The idea as well is as the presentation of the paper is very strong. This is a significant improvement over current methods of fMRI encoding. Making sure that voxel responses are anatomically correlated respects the realities of fMRI data in a way that treating voxels independently does not. It's also impressive how the implementation is fairly straightforward and as the experiments show, it works.

**Weaknesses:**

One weakness is that the results are demonstrated on a single dataset. Generalization across subjects is again, within the same dataset. I think the method would become an even better sell if it can be shown to generalize on other datasets, especially across sites. As that generalizability is a hard problem in the fMRI space.

The other weakness is in the ultimate usefulness, the fact that it can generalize to a new subject is great, but that still requires some data from that subject. From a practical standpoint that means you still will need to put the subject through a scanner. If we take that as an inevitability, what we'd like to see is that the only data we need is from some sort of standardized or just from the beginning of the scan - so that the subject can then be let go after a short amount of scanning. With the current set-up, (I assume) random portions of the new participant's data are being used for finetuning. That might not be practical in reality.

**Questions:**

See weaknesses.

---

> ### Author Response · Authors · 2025-11-23
> **Response to Reviewer vXEp**
>
> We thank the reviewer for their very thoughtful comments and feedback! Please see below for our responses.
>
> >**W1:_Results demonstrated only on a single dataset_**
>
> **Response:**  Thank you for raising this point! We agree that cross-dataset and cross-site generalization is an important next step for our work！ We focused on NSD because it is the most coomonly used large-scale naturalistic dataset with sufficient trial repetitions and SNR to support controlled voxelwise encoding and subject-adaptation experiments. Importantly, NRF’s continuous, coordinate-based formulation is explicitly designed to support cross-subject and cross-site generalization, and we view multi-site evaluation as an exciting next step.
>
> >**W2:_Practical usefulness: adaptation still requires data from the new subject_**
>
> **Response:** We thank the reviewer for this insightful point! We agree that minimizing the subject-specific data burden is crucial for real-world applicability! In our experiments, the fine-tuning requires only a very small number of images, meaning that the method effectively needs only a brief calibration scan rather than a substantial portion of the dataset. Although our paper samples these images randomly for experimental consistency, the key takeaway is that **very short, limited scanning is sufficient** for effective adaptation. We fully agree that moving toward standardized or fixed calibration protocols would further improve practicality! Designing a brief, shared stimulus set or even exploring adaptation from resting-state data is a compelling future direction.

---

### Official Review · Reviewer_vmAw · 2025-10-29

**Soundness:** 2
**Presentation:** 2
**Contribution:** 2
**Rating:** 2
**Confidence:** 5

**Summary:**

This work is part of a broader research effort to study voxelwise encoding models for visual stimuli. While prior vision encoding studies typically used linear mapping between image features and voxel activations of image stimuli. The current work advances the prior work by proposing a Neural Response Function (NRF) that takes both voxel location (x,y,z coordinates from a 3D volume) and image features as input, and uses an MLP(multi-layer perceptron) to predict fMRI brain activity. The authors claim that including voxel coordinates provides local spatial smoothing, capturing neighboring voxels, and an MNI space brings all subjects in common space possible for cross-subject predictions. Using the NRF approach, the evaluation focuses on comparing the individual subjects with baseline models such as linear regression and feature-weighted receptive field (fWRF) method for both brain encoding and brain decoding. Furthermore, the authors test cross-subject transfer: train on one subject and then fine-tuned on a new subject for adaptation. Overall, proposed NRF shows strong encoding and decoding performance under low-data regimes, provides an anatomically aware neural encoding model, generalizes both spatial smoothness and supports cross-subject alignment.

**Contributions:**

* *MNI space voxel coordinates for neural encoding:* The study uses both spatial coordinates (x,y,z) of each voxel and image stimuli representations from image encoder and use these representations in NRF to predict fMRI brain response to visual stimuli. Specifically, incorporating anatomical coordinates offers local spatial smoothing and facilitates cross-subject alignment, which is methodologically novel, relative to prior work that treats voxels as flattened 1D activations.
* *Comprehensive evaluation:* The study evaluates proposed NRF’s brain predictive performance on individual subjects and compares it against baseline models including linear regression and fWRF. For each encoding model, the authors report correlation and MSE, for decoding, they use image retrieval metrics (e.g., SSIM, PixCorr). Further, they compare cross-subject transfer by training one subject with varying data scales and fine-tuning on a new subject, quantifying new subject adaptation as a function of training steps and dataset size, with an emphasis on low-data regimes.

**Technical summary:**
This is primarily an empirical study, and its methodology involves the following components:
* *Mapping spatial coordinates:* The authors define a continuous function over MNI coordinates (common space for all subjects). For each voxel coordinate r =(x,y,z), the function computes a positional encoding ϕ(r) using fourier positional encoding method (which is popular in Transformer language models). The positional encoding vector is used in the NRF function during encoding to predict fMRI response.
* *Neural Response Function:* To train the NRF, the model takes two inputs: Image stimulus representations from image encoder and positional encoding features (ϕ(r)) for each voxel coordinate r. These vectors are concatenated and passed as input to MLP that predicts target fMRI response at spatial coordinate r. The model is trained end-to-end with a weighted sum of convex combination of mean square error and cosine similarity between the predicted response  and ground truth fMRI; optimised using Adam.
* *Cross-subject transfer:* The authors use a two-step strategy: (i) end-to-end fine-tuning- it involves fine-tuning a pretrained NRF model on a new subject's limited data and performing end-to-end fine-tuning. (ii) Voxel-wise ensembling: In this step,  for each voxel, they make predictions from K fine-tuned models for an image by learning voxel-specific weights for each fine-tuned model.

**Experimental design/evaluation:**
* *Subject-specific encoding:* The authors evaluate the capability of NRF by training subject-specific models and compare them with two baselines: linear regression and fWRF. To probe NRF under limited data samples, they train with 20, 40, 200, 400, 600, 800 and full images and report performance across four subjects.
* *New subject adaptation:* This analysis evaluates cross-subject transfer, where NRF trained on three subjects with 20, 200, and 800 training images, then test on the held-out fourth. For the fourth subject, the authors fine-tune the pretrained NRF and compare it to an NRF trained from scratch. They also perform voxel-wise ensembling: for each voxel, predictions from K fine-tuned models are combined using voxel-specific weights.
* *Probing anatomical awareness:* To test whether NRF really uses spatial continuity and cross-subject anatomy, the authors perturb the coordinates: they disrupt local smoothness by shuffling voxel locations within a subject, and shifting MNI coordinates across subjects to break the alignment. Performance drops under these perturbations, indicating both spatial continuity and anatomical alignment are crucial for better NRF performance.


**Main findings:**
According to the authors’ interpretation, the main findings are as follows:
* Under subject-specific encoding across different training data regimes, with only 200 training samples, NRF outperforms baselines trained on more than 800 images, indicating stronger performance in low-data regimes.
* During cross-subject transfer, NRF with fine-tuning + ensemble consistently outperformed NRF scratch, indicating the anatomically conditioned formulation enables efficient cross-subject transfer with limited target data while maintaining predictive accuracy.
* Shuffling local-spatial continuity and shifting MNI coordinates both reduces NRF performance, which implies NRF relies on local spatial continuity and cross-subject anatomical alignment. Overall, authors argue that NRF serves as a resolution-agnostic representation that can integrate data across scales and subjects.

**Strengths:**

I found this work to have the following strengths:
* *Clarity:* The manuscript introduction, dataset, and key methodological details are well written and well structured. The pipeline top panel of Figure 1 is easy to follow and clearly shows a pretrained NRF for a single subject. Later, the NRF formulation is described clearly combining an image-embedding vector with a positional encoding.  The model training and cross-subject transfer are also explained well, including how fine-tuning is performed on the pretrained NRF. The results section clearly reports subject-specific encoding performance, compares NRF with two baselines, applies the approach to cross-subject transfer, and then presents perturbation analyses that probe the proposed NRF.
* *Originality:* The idea of using common-space MNI coordinates to build positional encodings and combine with image features for handling local-spatial smoothing in a neural encoding model to predict fMRI is a simple idea that is well known in brain decoding but remains underexplored in brain encoding. Prior brain encoding studies typically use flattened 1D brain responses and learn a subject-specific ridge-regression model. In contrast, NRF provides a cross-subject model that supports subject adaptation and outperforms baselines in low-data regimes.
* *Significance:* This work is significant in that it contributes to a better understanding of the contribution of local-spatial smoothing and anatomical alignment are crucial for better encoding performance under low-data regimes. The comprehensive analysis on the NSD dataset shows that NRF serves as a general-purpose brain model that supports cross-subject transfer and can integrate data collected at different spatial resolutions.

**Weaknesses:**

From my perspective, the primary weaknesses of this study arise from the lack of comparison with prior literature, and limited evaluation:
* *Limited methodological novelty:* The paper claims that NRF is “the first anatomically aware encoding model to move beyond flattened voxels, learning a continuous mapping from images to brain responses in 3D space”. However, prior study, MindLLM (Qiu et al. 2025) already uses voxel coordinates with Fourier positional encoding for embedding vectors for each voxel coordinate (x,y,z). Given this, I recommend authors to soften the word “first claim” and clarify how NRF differs from prior work. Further, the authors ignore the prior study, and do not cite or compare against MindLLM.

Qiu et al. 2025, MindLLM: A Subject-Agnostic and Versatile Model for fMRI-to-Text Decoding, ICML-2025

* *Lack of motivation and position:* The authors argue that flattened neural 1D response vectors make them forced to train subject-specific models and it leads to poor data efficiency. However, this claim is not fully convincing. In general, the 3D volume can be mapped onto common space (e.g., MNI or fsaverage/fsaverage5/6), enabling multi-subject models and cross-subject transfer. Further, prior work does not ignore 3D spatial information as they use localizers to map language/Visual ROI voxels [Huth et al. 2016, Deniz et al. 2019]. I suggest authors clarify why conditioning on voxel coordinates within NRF is necessary beyond these standard options, and what extra benefit it provides (ideally with a direct comparison).

[Huth et al. 2016] Natural speech reveals the semantic maps that tile human cerebral cortex, Nature 2016

[Deniz et al. 2019] The representation of semantic information across human cerebral cortex during listening versus reading is invariant to stimulus modality, Journal of Neuroscience 2019

* *Inconsistency of evaluation metrics:* The dataset section 4.1 (line 263) states that the NRF model is trained and tested with model accuracy via R^2, while line 271 states that Pearson correlation and mean-squared error as voxel-level metrics. There are some results reported with Pearson correlation in Fig2a, Table1, and 2, while Fig2b shows prediction accuracy. Therefore it is unclear, which metrics are primary, and what they reported in results? I recommend authors to clarify the primary evaluation metric and use them consistently across Tables and Figs.
* *No statistical significance reports:* The paper reports no statistical significance tests (e.g., bootstrap testing with FDR correction or paired permutation) to determine whether model differences are significant. Without any significance tests, the performance of NRF over baselines should be considered as descriptive rather strong empirical evidence.
* *No standard error or standard deviation across subjects:* The results reported in Table 1-2 state that all metrics are calculated across 4 subjects, without accompanying measures of variability such as standard deviation or standard error. These metrics are critical for evaluating the robustness of the findings.
For a complete and detailed account of both major and minor issues, please refer to the “Questions” section.

**Questions:**

I would like to thank the authors for the interesting comparison between coordinate-conditioned NRF (leveraging local spatial smoothing via voxel coordinates) and linear regression models with flattened 1D brain responses for fMRI encoding during visual stimuli. However, there are several points that I believe require further attention/work. I have divided these into major issues, which should be prioritized, and minor ones, which should be addressed for a strong version of current work.

**Major Comments/Questions**
* *Motivation and positioning:* While cross-subject transfer and multi-subject models are possible with surface-level alignment (fsaverage surface). To strengthen the claims, I recommend authors to add: (i) multisubject models in average baseline, and (ii) compare NRF model in MNI space vs. multi-subject average baseline. If NRF still outperforms the fsaverage baseline, especially in low-data and cross-subject transfer settings, that would strengthen the local spatial smoothing claim and support the conclusions.
* *Statistical significance and clarity of Fig 2:* Fig2a reports subject-specific encoding model performance for two baselines (linear regression and fWRF) and proposed NRF. However, no statistical significance tests are provided. From Fig 2a, it is hard to conclude that NRF is significantly better than baselines. I suggest authors following
     * Please perform paired permutation/bootstrapped tests with Benjamini–Hochberg FDR correction (across voxels or per-subject summaries), report the adjusted p-values and effect sizes, and clearly indicate which model pairs are statistically significant.
     * Select voxels which are significant for each model and report percentage of significant voxels across subject-level summaries.
     * Repeat above two steps  while varying the number of training images and report significant results.
     * Also clarify what evaluation metric is reported in Fig 2b. There is an inconsistency of evaluation metrics mentioned in methodology that R^2 as accuracy and Peason correlation as voxel-level evaluation metric.
     * Similarly, no statistical significance tests are provided for Fig 4. From Fig 4, it is hard to conclude that NRF is significantly better than ROI shifted results for EBA and V1 regions.
* *Reporting variability:*  Please report standard deviations (or standard errors) alongside mean/median values to provide a clearer picture of variability in Table 1and 2. Further, fix the caption in each Table to state explicitly whether values are means (or medians) across the 4 subjects.
     * Table 3 shows transfer only for Subject 7. Please run the analysis for every target subject (leave-one-subject-out), and report mean ± SD (or SEM) per target subject as well as the overall average. A full source->target transfer matrix (heatmap) with uncertainty and FDR-corrected paired tests would make the results much more robust.
* *Clarity on fMRI-to-Image reconstruction:*  While authors report image reconstruction results in Table 2 and 3 with semantic-level metrics, and qualitative analysis of images in Fig 3 using MindEye2, there are no details about what information exactly is fed to MindEye2 from NRF model.
     * Please add a small description about fMRI-to-Image decoding
     * Also, in Table 2, NRF trails fWRF and MindSimulator on decoding. If decoding is not the paper’s strength, I recommend authors to reposition decoding as a diagnostic, highlighting where NRF adds value to the paper.
* *Why an MLP (NRF) instead of a linear/ridge model?*
     * The authors do not provide any justification why a nonlinear MLP is necessary once after concatenation of image features + positional encodings. A ridge regression as baseline on the same inputs would test whether the gains come from nonlinear interactions or simply from adding coordinates.

**Minor Comments/Typos**
While addressing the following points may not be critical to the paper’s core contributions, doing so would enhance the overall quality.
* I would appreciate authors to add little details for decoding setup.
* Clarify bottom panel Figure 1: It is unclear why both the pretrained NRF and the fine-tuned NRF are shown for Subject 1. Please state explicitly in the caption: Need better clarification in Figure 1 caption.
* Line 42, 46, 97: Follow proper citation format: e.g., “Downing et al. (2001) -> (Downing et al. 2001)”. Please correct all over the paper and use appropriate citation format: \citep{} or \cite{}.
* Line50: 2)Subject specific -> 2) Subject specific
* Line 103: Please add citation for this sentence: “Despite individual variability->”
* Line 152: define abbreviation: Implicit neural representations -> Implicit neural representations (INR)
* In Table captions, specify whether metrics are per-voxel median or mean over voxels and how subject aggregation is done.
* Figure 2b: please add evaluation metric on colorbar
* Since NSD provides pycortex flatmaps; please add 2D cortical flatmaps to complement Fig. 2b and make differences visible: Show absolute performance maps (e.g., voxel-wise Pearson r or R^2) for 200 vs. 800 images for each model (linear, fWRF, NRF) with the same color scale across panels. Current Fig 2b, the brain maps look similar across models.
* Move Fig. 3 to the appendix. Align Fig. 2 and Fig. 4 layouts. Use the same panel grid, and increase the font size of legends and axis-labels.

**General Advice**
The manuscript presents a Neural Response Function (NRF) that uses voxel coordinates for local-spatial smoothing and combining with image features to perform visual encoding on NSD dataset. Using NRF, the authors evaluate subject-specific models, low-data regimes, and cross-subject transfer. However, the current version lacks strong alignment baselines (ideally, an fsaverage/surface baseline), and shows metric inconsistencies (R^2 vs. Pearson/MSE), and reports results without any statistical significance testing. The decoding section is under-specified and NRF underperforms relative to baselines. Figures would benefit from reorganization and clearer labeling, and tables should include subject-level variability. Addressing these points and the above mentioned weaknesses and major comments would make the work stronger.

---

> ### Author Response · Authors · 2025-11-23
> **Response 1/4 to Reviewer vmAw**
>
> We thank the reviewer for their very thoughtful comments and feedback! Please see below for our responses.
>
> >**_W1: Limited methodological novelty_**
>
> **Response:** We thank the reviewer for raising this important point and for directing us to MindLLM (Qiu et al., 2025). We acknoledge that MindLLM also incorporates 3D spatial coordinate information in its modeling framework, and we have updated the Related Work section to more clearly articulate the conceptual and methodological differences between MindLLM and our approach. We have also adjusted the wording to avoid confusion in the revised PDF.
>
> Below, we clarify the distinction along two dimensions: the task setting and the role of spatial information.
>
> **1) Different problem settings: decoding vs. encoding.**
> MindLLM is designed for the **decoding** problem, generating text from measured fMRI activity. Its pipeline takes voxel responses as input and, and voxel coordinates are incorporated within a neuroscience-informed attention mechanism. In contrast, our work tackles the **encoding** problem, predicting neural responses from visual stimuli. This requires modeling the forward mapping from images to brain responses.
>
> **2) Different roles of spatial coordinates.**
> Although both approaches use voxel coordinates, their function is very different from ours. In MindLLM, the model must operate directly on the discrete voxel responses measured for each subject and at the fixed spatial resolution. The spatial coordinates therefore can only be used as additional features attached to each voxel token (e.g., Fourier-encoded positional embeddings inside the attention mechanism). hese embeddings help structure voxel–voxel interactions, but they **do not change the fact that MindLLM is defined only on the subject’s acquired voxel grid**.
> In contrast, NRF treats the brain as a continuous 3D structure rather than a fixed set of measured voxels. Spatial coordinates are not auxiliary features but define the domain of a continuous neural field in standardized anatomical space. This formulation enables resolution-agnostic querying, principled interpolation beyond the acquired voxel grid, and structured cross-subject alignment within a unified coordinate system. —capabilities that voxel-discrete decoder frameworks like MindLLM cannot support.
>
> **Summary:**
> MindLLM and NRF both acknowledge the importance of spatial information, but in **distinct ways** aligned with different objectives. MindLLM embeds coordinates to enhance the fMRI tokens for decoding within a voxel-discrete framework whereas NRF introduces a **continuous, anatomically aligned neural field** for the encoding task. We have added citation and discussion of MindLLM to the Related Work section.

---

> ### Author Response · Authors · 2025-11-23
> **Response 2/4 to Reviewer vmAw**
>
> >**_W2: Lack of motivation and position_**:
>
> **Response:** We thank the reviewer for the helpful suggestions and for pointing us to additional relevant work. We have added discussion of these works and clarified this distinction in the revised manuscript.
> We clarify this question along two dimensions:
>
> **1) Cross-subject transfer**
>
> We fully agree that cross-subject transfer is *possible* with flattened voxel
> representations, provided that each subject’s 3D fMRI volume is first mapped onto a
> common anatomical grid such as MNI or fsaverage. Our intention was not to imply that
> cross-subject modeling is impossible in this setting. Rather, the distinction we aim to
> highlight concerns the **processing burden and constraints** introduced by such
> Pipelines.
>
> Cross-subject models based on flattened voxel vectors require **volumetric or surface resampling**. This preprocessing step—whether mapping volumes into MNI space or projecting onto fsaverage surfaces—is computationally expensive, resolution-dependent, and may introduce interpolation artifacts. More importantly, it ties the encoding model to a
> fixed grid representation, reducing flexibility when working with different voxel
> resolutions or acquisition protocols.
>
> In contrast, **NRF avoids this requirement entirely**. Because NRF conditions its
> encoding function directly on each voxel’s *MNI coordinates*, cross-subject adaptation
> only requires providing the (x, y, z) location itself—**not resampling
> the full volume**. The model learns a continuous function over anatomical space, making
> cross-subject transfer more flexible, lightweight, and resolution-agnostic.
>
> Our motivation is therefore not that 3D coordinates are strictly required for
> cross-subject transfer, but to emphasize that **explicit coordinate conditioning offers a principled
> and efficient alternative** that improves data efficiency and simplifies multi-subject
> modeling without enforcing a fixed volumetric grid.
>
> **2) Relation to prior work on brain semantic mapping**
>
> We appreciate the reviewer’s pointer to the semantic–mapping literature. In these works
> (e.g., Huth et al., 2016; Deniz et al., 2019), voxelwise encoding models are trained on
> **flattened 1D response vectors**, treating each voxel as an independent output. The 3D
> organization of the cortex is then recovered **after training** by mapping the
> learned selectivities back onto the cortical surface or volume for visualization.
>
> Thus, while these studies leverage anatomical information during *analysis*—for example,
> using ROI localizers or surface maps—the encoding models themselves **do not integrate
> 3D spatial structure during learning**. Spatial organization is reconstructed *post hoc*,
> and the models do not use voxel coordinates as part of the predictive function.
>
> In contrast, **NRF incorporates continuous 3D voxel coordinates directly into the
> encoding architecture**. By conditioning the model on anatomical location, NRF injects
> spatial structure into the learning process, enabling it to exploit spatial smoothness
> *during* training rather than only in downstream visualization. This design choice yields
> improved performance in the low-data regime and supports efficient cross-subject
> adaptation, as shown in our experiments. Importantly, **the findings from semantic–mapping studies motivate our approach**: they show that nearby voxels exhibit smoothly varying semantic selectivity. NRF
> operationalizes this principle by embedding anatomical coordinates into the
> encoding model itself, rather than using them solely for post hoc interpretation.

---

> ### Author Response · Authors · 2025-11-23
> **Response 3/4 to Reviewer vmAw**
>
> >**_W3 and  Q2: Inconsistency of evaluation metrics_**
>
> **Response:**  We thank the reviewer for raising this question and apologize for the confusion. The metric used throughout the paper is the Pearson correlation coefficient *R*. In the Methods section, the reference to “prediction accuracy” should also state R, consistent with the definition provided in Section 4.2 (“Evaluation Metrics”). We’ve corrected this typo in the revised manuscript. For Fig 2b correlation we are repeating the Pearson correlation coefficient *R*. We've revised the figure.
>
> >**_W4 and Q2: statistical significance reports_**
>
> **Response:**  We thank the reviewer for the detailed suggestions regarding statistical validation and clarity of Fig. 2a and Fig 4. Following the reviewer’s advice, we performed the full set of statistical significance analyses requested. We summarize the results below and include complete per-subject tables and voxel-level significance results in the **Appendix A.5**. The corresponding significance analysis for the ablation experiments in Fig. 4 is also included in **Appendix A.5**.
>
> **(1) Paired significance tests for Fig. 2a**
> For each subject and each training-image setting, we performed:
>
> - paired permutation tests (10000 permutations),
> - with Benjamini–Hochberg FDR correction (across voxels within each subject),
> - and reported effect sizes (Cohen’s *d*) and adjusted *p*-values.
>
> The table below summarizes the **mean ± SD effect size across subjects**, the **range of FDR-adjusted p-values**, and the **number of subjects with significant differences** for each comparison.
>
> | Training images | d (NRF vs fWRF) | p_FDR | # Sig | d(NRF vs Linear) | p_FDR) | # Sig |
> |-----------:|----------------------|----------------------|-------|------------------------|----------------------|-------|
> | 20         | 0.058 ± 0.223       | [0.000, 0.126]       | 3/4   | 0.308 ± 0.129          | [0.000, 0.000]       | 4/4   |
> | 60         | 0.058 ± 0.118       | [0.000, 0.000]       | 4/4   | 0.593 ± 0.075          | [0.000, 0.000]       | 4/4   |
> | 200        | 0.175 ± 0.151       | [0.000, 0.000]       | 4/4   | 0.638 ± 0.114          | [0.000, 0.000]       | 4/4   |
> | 400        | 0.243 ± 0.070       | [0.000, 0.000]       | 4/4   | 0.598 ± 0.070          | [0.000, 0.000]       | 4/4   |
> | 600        | 0.240 ± 0.110       | [0.000, 0.000]       | 4/4   | 0.543 ± 0.096          | [0.000, 0.000]       | 4/4   |
> | 800        | 0.260 ± 0.171       | [0.000, 0.0002]      | 4/4   | 0.483 ± 0.132          | [0.000, 0.000]       | 4/4   |
> | Full       | 0.175 ± 0.149       | [0.000, 0.000]       | 4/4   | 0.323 ± 0.183          | [0.000, 0.000]       | 4/4   |
>
> Across all training conditions, **NRF shows statistically significant improvement over both fWRF and the Linear baseline**, with medium-to-large effect sizes.
>
> **(2) Percentage of significant voxels across models for Fig. 2a**
> As requested, we computed the percentage of voxels significantly predicted above chance (FDR < 0.05). Below we report the **mean ± SD** across subjects.
>
> | Training Images | NRF (%)            | fWRF (%)         | Linear (%)       |
> |----------------:|--------------------|------------------|------------------|
> | 20              | **64.61 ± 7.22**  | 58.22 ± 8.70     | 49.86 ± 8.98     |
> | 60              | **77.14 ± 5.61**  | 72.46 ± 5.41     | 60.80 ± 7.23     |
> | 200             | **84.58 ± 4.80**  | 81.21 ± 5.58     | 74.71 ± 5.48     |
> | 400             | **88.57 ± 3.20**  | 85.49 ± 3.92     | 81.40 ± 4.47     |
> | 600             | **88.85 ± 4.20**  | 87.36 ± 3.94     | 84.75 ± 5.60     |
> | 800             | **89.85 ± 4.02**  | 88.55 ± 3.92     | 86.09 ± 3.91     |
> | Full            | **93.03 ± 2.48**  | 91.36 ± 2.09     | 92.52 ± 2.39     |
>
> NRF consistently predicts a larger fraction of voxels above-chance than both baselines, especially in low-data settings.
>
>
>
> >**_W5 and Q3: variability across subjects_**
>
> **Response:**  We thank the reviewer for highlighting the importance of reporting variability. We have now included tables reporting the **mean ± SEM of the subject-level medians**  in the revised appendix.  Specifically, **Appendix A.4 (Table 5)** provides the variability corresponding to Table 1, and **Appendix A.6 (Table 20)** provides the variability for Table 2. Per-subject results are also included in these sections. We have additionally revised the table captions to explicitly state how metrics are aggregated. For Table 1, voxel-level metrics are reported as the mean of the per-subject medians across the four subjects. For Table 2 (new-subject adaptation), the voxel-level metric represents the median across all voxels of the target subject (S7).

---

> ### Author Response · Authors · 2025-11-23
> **Response 4/4 to Reviewer vmAw**
>
> >**Q1: _fsaverage baseline_**
>
> We thank the reviewer for this helpful suggestion. We address the points raised as follows:
> - **Cross-subject transfer vs. fsaverage alignment.**
> We agree that cross-subject and multi-subject modeling can be performed using surface-based alignment methods such as fsaverage. As noted in our response to **W1**, the goal of NRF is to enable cross-subject modeling **without requiring explicit volumetric resampling or surface projection** of each subject’s fMRI volume. Importantly, NRF does not exclude the use of aligned data—its coordinate-conditioned formulation can operate on MNI coordinates, fsaverage surface coordinates, or any other standardized space. Alignment affects the coordinate system supplied to the model, whereas NRF’s benefits arise from how the coordinates are used, not from the choice of anatomical template.
>
>
> - **fsaverage baseline is not directly comparable to the current NRF.**
>   fsaverage-based methods operate on a **2D cortical surface representation**, whereas NRF is defined over the **full 3D volumetric space**. These representations differ fundamentally in anatomical support (surface vs. volume), geometric structure, and spatial topology. A controlled comparison would therefore require designing a **surface-based variant of NRF**, which is beyond the scope of the present work. We acknowledge this as a valuable direction for future extensions!
>
> - **NRF’s advantages do not depend on whether alignment is applied.**
> The reviewer asks whether NRF’s improvements would persist if an fsaverage multi-subject baseline were added. The key point is that NRF’s gains come from the **continuous neural-field formulation**, which imposes local spatial smoothness during training. This mechanism is **independent of the alignment method**. NRF would retain the same advantages whether voxel coordinates come from raw MNI alignment, fsaverage surface alignment, or any other standardized mapping.
>
>
> >**Q4: _Details on semantic eval_**
>
> We thank the reviewer for this helpful suggestion. We’ve added clarification in the revised PDF. Given an input image, we first predict its fMRI response using NRF; the predicted response is then fed into MindEye2, which reconstructs the corresponding visual stimulus. We compare these reconstructions against the ground-truth images presented during data collection.
> In the **Response to All Reviewers**,  we have included a discussion clarifying the interpretation of the semantic evaluation metric. We have also added a subsection in **Appendix A.3** that further analyzes this issue and clarifying the limitations of using such semantic metrics for comparing neural encoding models.
>
> >**Q5: _Why an MLP (NRF) instead of a linear/ridge model_**
>
> We thank the reviewer for raising this insightful point. The key distinction is that in a standard regression framework, **each voxel is modeled independently**, whereas NRF models the **entire brain’s response field as a single continuous function** defined over 3D space. Because regression treats voxels as separate prediction targets, simply adding coordinate features does not allow the model to capture spatial smoothness or shared structure across voxels in the way that NRF does.
> To verify this, we added a baseline comparing:
>
> - **Image embedding → voxelwise regression**
> - **Image embedding + voxel coordinates (positional encoding) → voxelwise regression**
>
> In both cases, a separate regression model is fit for each voxel. The results below report the **median voxel prediction accuracy** per subject:
>
> | Model                                           | S1    | S2    | S5    | S7    |
> |------------------------------------------------|-------|-------|-------|-------|
> | Image embedding regression                      | 0.29574| 0.30973| 0.40540| **0.26691** |
> | Image embedding + PE regression | **0.29576** | **0.30989** | **0.40564**|  0.26690 |
>
> We performed paired voxel-level significance tests (permutation test + BH–FDR). The effect sizes and adjusted p-values are shown below for Image embedding regression vs Image embedding + PE regression.
> | Subject | Cohen’s d | p_FDR | Sig.? |
> |------:|-----------:|--------:|:------|
> | S1    | -0.0568    | 0.0000 | Yes   |
> | S2    | -0.0798    | 0.0000 | Yes   |
> | S5    | -0.0724    | 0.0000 | Yes   |
> | S7    |  0.0058    | 0.5166 | No    |
>
> Across subjects, effect sizes are extremely small and inconsistent in direction. This shows that simply **adding positional encodings to voxelwise regression does not meaningfully improve prediction accuracy**. The benefit does not come from providing coordinates alone, but from jointly modeling all voxels through a **coordinate-conditioned neural field** that learns spatial structure, something independent regression models cannot exploit.
>
> >**_Minor Comments/Typos_**
>
> We thank the reviewer for raising these points! We've addressed them in the revised PDF.

---

### Official Review · Reviewer_mipv · 2025-11-01

**Soundness:** 3
**Presentation:** 3
**Contribution:** 3
**Rating:** 6
**Confidence:** 4

**Summary:**

This paper introduces the Neural Response Function (NRF), a novel neural encoding model that represents fMRI activity as a continuous function over 3D anatomical space. Unlike conventional models that flatten brain data into 1D vectors, NRF is an implicit neural representation that takes a visual stimulus and a standardized MNI coordinate (x, y, z) as input to predict the response at that specific location. This anatomically-aware formulation allows the model to leverage the spatial smoothness of fMRI data and enables efficient cross-subject transfer, as the MNI coordinate system is shared across individuals.

**Strengths:**

1. Novel and Elegant Formulation: The core contribution, modeling brain responses as a continuous, coordinate-based implicit function, is a novel and elegant departure from the standard "grid-locked" voxel-wise approach.

2. Solves a Key Practical Problem (Low Data): The model's strongest results are in the low-data regime, where it substantially outperforms baselines. This is a highly relevant contribution, as most fMRI studies outside of massive public datasets are data-scarce. The ability to exploit spatial priors (smoothness) clearly improves data efficiency.

3. Effective Cross-Subject Transfer: The paper successfully demonstrates a practical framework for cross-subject adaptation. By grounding the model in MNI space, it provides a principled way to transfer knowledge, and the finetune-ensemble strategy yields impressive performance with minimal data from a new subject.

4. Strong Empirical Validation: The probing experiments are commendable. By intentionally breaking the model's assumptions (shuffling coordinates) and observing the expected performance drop, the authors provide strong evidence that the model's gains are in fact due to its leveraging of spatial smoothness and anatomical alignment.

**Weaknesses:**

- Unaddressed Computational Cost: The per-coordinate query architecture seems significantly less scalable than standard single-pass encoders. The paper provides no analysis of this computational overhead or its implications for full-brain, high-resolution inference.

- Incomplete Related Work Context: The paper emphasizes its cross-subject adaptation. However, the related work section overlooks that many recent neural decoding models (e.g., 'MindBridge' [1]) also incorporate encoding mechanisms and explicitly address the challenge of adapting to new subjects. The Related Work section would be more comprehensive with a discussion of these approaches.

**Questions:**

1. Performance in Full-Data Regime: Why does the significant performance gap observed in the low-data regime (Fig 2a) largely disappear in the full-data setting (Table 1)? Does the strong spatial-smoothness prior perhaps act as a regularizer that helps when data is scarce but hinders the model from capturing fine-grained, non-smooth voxel-specific details when data is abundant?

2. Reliability of Semantic Metrics: Given that the fWRF baseline outperforms the "Measured fMRI" ground truth on several semantic metrics, how should we interpret these results? Does this not suggest the MindEye2 decoder is (over)fitted to fWRF-like distributions, making this entire family of metrics unsuitable for comparing these encoders?

I will consider raising my score if all my concerns are well addressed by the authors.

---

> ### Author Response · Authors · 2025-11-23
> **Response to Reviewer mipv**
>
> We thank the reviewer for their very thoughtful comments and feedback! Please see below for our responses.
>
> >**_W1: Unaddressed Computational Cost compared to standard single-pass encoders_**
>
> **Response:** We thank the reviewer for raising this point! We’d like ro clarify that both NRF and standard single-pass encoders have **the same linear computation complexity scaling in the number of voxels**. Because all voxels for a subject share the same NRF model, the network supports batched evaluation of $(x,y,z)$ coordinates.  Let the per-voxel cost of evaluating the shared MLP be $O(C_{\text{NRF}})$, where $C_{\text{NRF}}$ is a constant. Querying $N$ voxels therefore has total complexity $O(N \cdot C_{\text{NRF}})$. For a standard single-pass voxel-wise encoder such as a linear regression model, prediction corresponds to computing $Wz$, $W$ is the weight matrix with shape ($N$, $d$), where $d$ is the dimensionality of the image feature. This computation has cost $O(N \cdot d)$, Thus, **both approaches scale linearly** in the number of voxels $N$, and increasing spatial resolution does not change this asymptotic comparison.
>
> Furthremore, because the NRF MLP isshared across all voxels, the **model size does not grow with the number of voxels**. In contrast, a voxel-wise linear model contains $N \cdot d$ parameters. Consequently, NRF is **more scalable in model capacity and memory footprint**, making it advantageous for high-resolution or full-brain inference.
>
> >***W2***: **_Incomplete Related Work Context_**
>
> **Response:** We thank the reviewer for highlighting this important point! In the revised PDF, we have expanded the Related Work section to include a more comprehensive discussion of recent neural decoding frameworks, including *MindBridge* that address cross-subject adaptation.
>
> >***Q1***: **_Performance in Full-Data Regime_**
>
> **Response:** We thank the reviewer for raising this point! We'd like to clarify that the narrowing of the performance gap in the full-data regime is expected and does not indicate that NRF’s spatial prior suppresses meaningful high-frequency voxel-specific structure.
> - **Voxel responses remain spatially smooth even with abundant data.**
>   The inherent organization of visual cortex produces locally smooth response patterns: nearby voxels within ROIs such as FFA or PPA exhibit highly similar semantic tuning. This biological smoothness does not disappear with additional training samples. NRF’s spatial conditioning therefore reflects genuine biological structure rather than oversmoothing or removing high-frequency signal.
>
> - **NRF’s advantage in low-data settings comes from reduced sample complexity.**
>   When training data is scarce, NRF can share information across anatomically similar voxels, substantially reducing the number of samples required to learn voxel-level mappings. Baseline models learn each voxel independently and therefore struggle in this regime. With enough training data, baselines eventually receive sufficient voxel-specific supervision to catch up—naturally reducing the gap.
>
> - **The noise ceiling constrains accuracy in the full-data regime.**
>   In the full-data setting, many voxels prediction accuracy approach their noise ceiling. Once this upper bound is reached, additional model capacity yields diminishing returns. As a result, prediction performance across models becomes closer.
>
> In summary, the gap closes because (i) baseline models improve when voxel-specific data becomes sufficient, and (ii) the data’s noise ceiling constrains further gains—not because NRF’s spatial prior hinders the learning of fine-grained structure.
>
> >***Q2***: **_Reliability of Semantic Metrics_**
>
> **Response:** We thank the reviewer for raising this important point! We agree that the behavior of the semantic metrics warrants careful interpretation. In the **Response to All Reviewers**, we have included a detailed discussion clarifying this point. We have also added a subsection in **Appendix A.3** that further analyzes this issue and clarifying the limitations of using such semantic metrics for comparing neural encoding models.

---

### Author Response · Authors · 2025-11-23
**Response to all Reviewers: Clarifying the Interpretation of the semantic level metrics**

We appreciate the reviewers’ comments regarding the semantic-level reconstruction metrics!

These semantic metrics (e.g., SSIM, AlexNet, Inception, CLIP, SwAV) are computed using an external fMRI-to-image decoder (MindEye2). Because this decoder is itself a trained network with its own inductive biases, its feature space can sometimes align more closely with the **statistical structure of predicted responses** than with the true measured fMRI signals. This can lead to counterintuitive cases where a baseline model (e.g., fWRF) scores higher than ground-truth fMRI on certain semantic metrics.
Such outcomes reflect **decoder bias**, not superiority of the predicted responses. Therefore, these reconstruction metrics should not be interpreted as primary indicators of encoding accuracy. they serve as **diagnostic visualizations** showing whether predicted responses contain enough structure to support plausible reconstructions.
All quantitative comparisons and statistical conclusions in our paper rely exclusively on **voxel-level encoding metrics** (e.g., Pearson correlation, MSE), which directly measure alignment with ground-truth neural responses and are not affected by decoder bias. We have added a discussion in **Appendix A.3** to clarify this point.

---

### Author Response · Authors · 2025-12-02
**Final remark**

We sincerely thank the reviewers and the AC for their careful evaluation and constructive feedback. We are grateful that multiple reviewers recognized the following key strengths of our work:

- **Novel continuous, coordinate-conditioned encoding formulation** *(mipv, vmAw, vUPt, vXEp)*
- **Strong advantages in the low-data regime, addressing a key practical problem** *(mipv, vmAw, vXEp)*
- **Effective and practical cross-subject adaptation** *(mipv, vmAw, vUPt)*
- **Probing experiments to validate the source of gains** *(mipv)*
- **Clear presentation and strong anatomical grounding** *(vmAw, vXEp)*

---
We summarize the main concerns raised by the reviewers and how they were addressed during rebuttal.

**1. Novelty vs. MindLLM** *(vmAw)*

*Concern: MindLLM already incorporates voxel coordinates with Fourier positional encodings*

**Response:** We clarified that MindLLM embeds coordinates as auxiliary features within a voxel-discrete **decoding** framework, whereas NRF introduces a **continuous, anatomically aligned neural field for the encoding task**. We added discussion of MindLLM and clarified the distinctions in the revised **Related Work** section.

---

**2. Motivation vs. Existing Cross-Subject Transfer Pipelines**  *(vmAw)*

*Concern: Cross-subject transfer can already be achieved by resampling and aligning flattened voxel representations.*

**Response:** We clarified that NRF’s contribution is not enabling transfer per se, but providing a **continuous, coordinate-conditioned formulation that eliminates volumetric/surface resampling**, avoids resolution dependence, and enables **lightweight, flexible, resolution-agnostic adaptation**.

---

**3. Motivation vs. Prior Brain Semantic Mapping Work**  *(vmAw)*

*Concern: Prior semantic-mapping studies already leverage 3D spatial information.*

**Response:** We clarified that prior work recovers spatial structure only **post hoc after training** for analysis purposes, whereas NRF **injects continuous 3D structure directly into the encoding function during learning**, enabling it to exploit spatial smoothness as a training-time prior, improving **data efficiency and cross-subject adaptation**.

---
**4. Statistical Significance Testing and Variability Reporting** *(vmAw)*

*Concern: No statistical significance tests or subject-level variability were originally reported.*

**Response:** We performed paired permutation tests with BH–FDR correction, reported Cohen’s d and percentages of significant voxels **Appendix A.5**. We reported mean ± SEM across subjects **(Appendix A.4, Table 5; Appendix A.6, Table 20)**. Per-subject results are also included.

---

**5. Inconsistency of Evaluation Metrics (R vs. R^2)**  *(vmAw)*

*Concern:* Pearson correlation R and  R^2 were both mentioned in the eval metrics.

**Response:** We clarified that Pearson correlation R is the primary encoding metric, it was an typo and has been corrected.

---
**6. Missing surface-alignment baseline** *(vmAw)*

*Concern:* No fsaverage or surface-based multi-subject baseline was included.

**Response:** We clarified that **fsaverage operates on 2D cortical surfaces**, whereas **NRF is defined in full 3D volumetric space**, making direct comparison non-trivial. We explicitly acknowledge this as **important future work (surface-based NRF variant)**.

---

**7. Encoding Model Performance vs. Prior Work** *(vUPt)*

*Concern:* The reported performance appeared low compared to previous studies.

**Response:** We clarified that this discrepancy arose from **different evaluation metrics and reporting conventions**. We re-evaluated NRF under the **same 10-fold cross-validation protocol** as [1] and reported **explained variance under matched settings**, showing that NRF is **competitive with and in several cases outperforms CLIP- and DINO-based transformer encoders**. We also clarified how **median-only vs. Q1–Q3** and **r vs. explained variance** caused misleading comparisons. The voxelwise prediction distribution plot(Q1–Q3) are included in the **Appendix A4 Figure.5** .

[1] Adeli, H., Sun, M., & Kriegeskorte, N. (2025). *Transformer brain encoders explain human high-level visual responses*. arXiv:2505.17329.

---

**8. Validity of Semantic Evaluation Metrics** *(vmAw, vUPt, mipv)*

*Concern:* The interpretation and validity of semantic reconstruction metrics were unclear.

**Response:** We clarified the full reconstruction pipeline and explicitly stated that **semantic metrics are used only as diagnostic tools**, not to support primary model comparisons. We included these metrics to provide **additional qualitative insight** into the behavior of the generated or selected images, following the evaluation format adopted in recent prior work, including *MindSimulator* [2] (ICLR 2025). Additional discussion was added in **Appendix A.3**.

[2] Bao et al. (2025). *MindSimulator: Exploring Brain Concept Localization via Synthetic fMRI*. ICLR 2025. arXiv:2503.02351.

---

### Meta-Review · Area_Chair_8kFh · 2026-01-07

**Summary:**

The paper proposes Neural Response Functions (NRF), a *continuous, coordinate-conditioned* fMRI encoding model that predicts the brain response at an arbitrary 3D location by taking a visual stimulus and a queried coordinate (in standardized MNI space) rather than outputting a fixed voxel grid, enabling resolution-agnostic inference and facilitating cross-subject transfer via shared anatomical coordinates. NRF is implemented as an implicit function $\Phi(M, x) = P(G(M), \gamma(x))$ combining an image feature extractor $G$, Fourier-feature positional encoding $\gamma$, and a predictor network $P$, trained by sampling voxel coordinates and optimizing regression losses. The authors also present a practical *new-subject adaptation* strategy (fine-tuning plus a voxel-wise ensembling scheme) and evaluate on the Natural Scenes Dataset (NSD) in visual cortex, reporting voxel-wise prediction accuracy and additional semantic-level decoding probes using predicted responses as input to an fMRI-to-image reconstruction model, alongside demonstrations of querying NRF on finer grids than the acquisition resolution.

**Reviewer Concerns:**

The main concerns centered on (i) novelty/positioning relative to closely related coordinate-aware models (e.g., MindLLM), (ii) missing or non-trivial alignment baselines (e.g., surface-based fsaverage-style comparisons), (iii) metric/reporting inconsistencies and lack of significance/variability reporting, (iv) perceived underperformance versus stronger contemporary encoders under common evaluation protocols, and (v) interpretability/validity of semantic reconstruction metrics.

The rebuttal is strong and directly addresses most high-impact issues: it clarifies the conceptual distinction from voxel-discrete coordinate-augmented approaches, adds paired permutation tests with BH–FDR correction and effect sizes, resolves the R vs. R² inconsistency, re-evaluates performance under matched cross-validation/reporting conventions, and explicitly reframes semantic metrics as diagnostic rather than primary evidence.

Remaining gaps are mostly about external validation and baselines (single-dataset evaluation; lack of a true surface-alignment comparator) and whether NRF’s performance remains competitive against state-of-the-art encoders that incorporate retinotopy/pRF structure.

**Reviewer Scores:**

The reviews are polarized: two rejects, one marginally above threshold, and one clear accept. I read the authors’ responses to the raised concerns and followed the ensuing discussion. While it is difficult to know whether the negative reviewers would have revised their scores, I find the core idea compelling; with the rebuttal-driven clarifications and added statistical rigor, the work reads as above-threshold overall.

---

### Decision · Program_Chairs · 2026-01-26

Accept (Poster)